

# Evaluation of single-band snow patch mapping using high resolution microwave remote sensing: an application to the Maritime Antarctic

Carla Mora[1], Juan Javier Jímenez [1,2], Pedro Pina[3], João Catalão[4], Gonçalo Vieira[1]

[1] CEG/IGOT, Universidade de Lisboa, Lisbon, Portugal
[2] Department of Physics and Mathematics, University of Alcalá de Henares, Alcalá de Henares, Spain
[3] CERENA/IST, Universidade de Lisboa, Lisbon, Portugal
[4] IDL/FCUL, Universidade de Lisboa, Lisbon, Portugal

*Correspondence to*: Carla Mora (carlamora@campus.ul.pt)

**Abstract.** Snow patch distribution and snow melt patterns during the summer are important controls for terrestrial ecosystems, permafrost and active layer, as well as for infrastructure access and management in the Maritime Antarctic. The mountainous terrain of the Maritime Antarctic and relatively small extent of the ice-free areas generate complex mosaics of numerous small snow-patches, ranging from tens to hundreds of meters in extension. These can only be accurately mapped using high resolution remote sensing sensors. However, the extremely high number of days with cloud cover limits the application of optical sensors from satellites, which have provided only sporadic snapshots in the Maritime Antarctic, limiting its use for monitoring purposes. In this paper we evaluate the application of Radar scenes from TerraSAR-X obtained in High Resolution SpotLight mode for mapping snow patches at a test area in Fildes Peninsula (King George Island, South Shetlands). Field analysis of the snow conditions, such as snow patch mapping and characterization of snow stratigraphy was conducted at the time of image acquisition in 12 and 13 January 2012. Snow was wet in all studied snow patches, with coarse-grain and rounded crystals showing advanced melting. Ice-layers were frequent in the snow pack. Two TerraSAR-X scenes in HH and VV polarization modes were analysed, with the former showing the best results in discrimination between wet-snow, lake water and bare soil. However, significant overlap in the backscattering signal was found. Average wet snow backscattering was -18.0 dB in HH mode, with water showing -21.1 dB and bare soil showing -11.9 dB. Single band pixel-based and object-oriented image classification methods were used to assess the classification potential of TerraSAR-X SpotLight imagery. The best results were obtained with an object-oriented approach using a watershed-based segmentation with a SVM classifier, with an overall accuracy of 92% and Kappa of 0.88. The main limitation was the west to northwest facing snow patches, which showed significant error an issue probably related to artefacts from the geometry of satellite imagery acquisition. The results show that TerraSAR-X in spotlight mode provides extremely high quality imagery for mapping wet snow and snow melt in the Maritime Antarctic. The classification procedure that we propose is a simple method and can easily be implemented in operational mode if a good digital elevation model is available.





## 1 Introduction

In mountainous terrain snow melt patterns during the summer are difficult to map accurately, especially when a high spatial resolution is necessary (e.g. < 5 m). High resolution satellite optical imagery is expensive, shows a large revisiting time and is only effective during the day and in cloud free conditions. Recently, Unmanned Aerial Vehicles have shown to provide

efficient snow mapping results at a low cost (Bühler et al., 2016) but they are still limited by the need of access to the survey area, as well as by meteorological conditions. In remote locations such as the Maritime Antarctic, with high cloudiness (ci 90% of the days show cloud cover) and affected by the continuous passage of polar frontal systems, more robust approaches are needed for monitoring snow melt over large areas.

Late lying snow patches are known to generate local influences on the ground thermal regime and on moisture availability,

thus being of major significance for geomorphic processes, ecosystems, and for permafrost distribution, especially in the discontinuous permafrost zone (Green and Pickering, 2009). Recent observations in the Maritime Antarctic indicate that snow patches play a key role in keeping the ground cooler during the summer, being determinant for the presence of permafrost at sites where, without summer snow, it would not occur (Vieira et al., 2010). The influence of snow patches on the geomorphological dynamics gave rise to the use of nivation as an overarching term for the complex set of geomorphic

processes acting in the vicinity of late-lying and perennial snow patches, with snow being their main driver (Thorn, 1988). Recurring nivation processes in the same location have been identified as responsible for the increased erosion and for the development of concavities, named nivation hollows. Snow is also a major ecological factor especially since it controls moisture availability during the warm season, but also because snow traps wind transported particles that are deposited in snow patches, allowing for a better development of the vegetation (Brown and Ward, 1996; Erickson and Williams, 2005; Hiemstra

et al., 2006; Green and Pickering, 2009). Snow also plays a major role in the distribution of lichen communities, inhibiting the development of *Usnea sp*. dominated formations (Vieira et al. 2014).

SAR and ASAR imagery, e.g. from ERS, Envisat (C-band) and TerraSAR-X are widely used to characterize snow packs and snow cover (Shi and Dozier, 1997; Baghdadi et al., 1999; Bernier et al., 1999; Nagler and Rott, 2000; Rees and Steel, 2001; Magagi and Bernier, 2003; Vogt and Braun, 2004; Longépé et al., 2009; Falk et al., 2016). Most applications have been

developed for regional scale mapping, but for high resolution approaches they lack quality. Despite the wide application of C-band imagery, Baghdadi et al. (1997) and Koskinen and Pulliainen (1997) have shown that wet snow and snow free terrain may not be possible to distinguish in some types of surfaces or in particular local incidence angles. Mora et al. (2013) tested ENVISAT ASAR C-band imagery at 12 m pixel resolution for mapping snow cover in Deception Island and found that the imagery is only useful at the regional scale and useless for snow patch mapping. In fact, according to some authors, X-band

imagery is preferable to detect wet snow (Shi and Dozier, 1995; Strozzi et al., 1998; Strozzi et al., 1999). It shows a limited penetration capacity in snow and is much more sensitive than other bands to the surficial snow pack (0 to 15 cm), allowing to evaluate the snow electromagnetic response in a simplified scheme when compared with C band (Wiessman and Mätzler, 1999; Rott et al., 2013).



TerraSAR-X acquisitions in Spotlight mode show *ci.* 1 m resolution and therefore are potentially a good source for very detailed snow mapping. The DLR satellite shows a short revisit time (11 days) and an improved radiometric and geometric resolution, which are key factors to detect the evolution of the snow cover, especially during snow melt when changing moisture content influences the backscattering signal. TerraSAR-X imagery is frequently used for interferometric applications

(Venkataraman and Rao, 2005; Alia et al., 2015; Barboux et al., 2015; Betbeder et al., 2015; Reis et al., 2015), glaciology (Braun, 2001; König et al., 2001; Rott et al., 2011; Schubert et al., 2013) and also for snow mapping (Baghdadi, et al, 1997; Malnes and Gunerissen, 2002; Malnes and Gunerissen, 2003; Venkataraman et al., 2008; Falk et al., 2016), but mostly using the coarser resolution StripMap mode. Most research focus on the retrieval of snow water equivalent (SWE) and not so much on the detailed mapping snow extent and melt patterns, topics which are very relevant to the geocryological community. In

fact, research on high spatial resolution mapping using microwave imagery is rarely present in the literature. Malnes et al. (2014) tested the use of TerraSAR-X SpotLight mode, VV-polarization imagery for SWE retrieval in Svalbard using ground truth data obtained along transects, but in order to reduce speckle noise, the authors used a 10 m pixel resolution, thus loosing resolution. They have found a good capacity for SWE estimation in dry snow, but in wet-snow, due to the complete absorbance of the radar signal in the top layers, the procedure did not work.

Climate scenarios indicate that the recent warming in the Antarctic Peninsula will be followed by an increase in precipitation and possibly in snow fall (Thomas, et al., 2008; Steig, et al., 2009; Winkelmann et al., 2012; Barrand et al., 2013). The significance of these changes for the geomorphological and ecological dynamics of the ice-free areas has not been yet evaluated. However, since the Western Antarctic Peninsula and especially the South Shetlands, show mean annual temperatures just slightly below 0 ºC at sea-level, the region will most probably suffer important effects, particularly in snow

cover and on the distribution and extent of late-lying snow patches. Mapping of the later and monitoring melting patterns, as well as interannual changes is therefore key for evaluating the changes in the ice-free environment, e.g. for permafrost, nivation and ecological research, but also for research infrastructure management.

This paper deals with evaluating the potential of TERRASAR-X (X-band) imagery acquired in SpotLight mode, to map summer snow patch distribution with a spatial resolution close to 1 m in the Maritime Antarctic. Spatial monitoring of snow

cover and snow melt has proven to be a very difficult task in the region (Mora et al., 2013; de Pablo et al., 2016) and the methodology proposed here aims at bridging this gap and at being implemented in operational mode to be made available to the terrestrial ecosystems and permafrost research community working on the Western Antarctic Peninsula. For the purpose of testing and validating, we have selected a field site in Fildes Peninsula (King George Island, South Shetlands archipelago).

## 2 Test site

The Meseta Norte is a mesa-like relief in the northeast part of Fildes Peninsula, King George Island (KGI), located in the South Shetlands, off the northern tip of the Antarctic Peninsula (Fig. 1). KGI is the largest island in the archipelago and about 90% of its surface (1,250 km$^2$) is glaciated, with Fildes Peninsula (62º 12′ S, 58º 58′W) being one of the largest ice-free areas of the



South Shetlands with 29 km² (Peter et al., 2008). Landforms are dominated by two high structural volcanic platforms (Meseta Sul, 167 m.a.s.l. at Promontório Schenke; and Meseta Norte, 155 m.a.s.l. at Cerro San Francisco). Low lying planation surfaces occur between and around them, especially in the northern part of Fildes Peninsula, with altitudes below 50 m.a.s.l.

The study site is located in the Meseta Norte, a plateau bounded by steep slopes with a slightly depressed central area at 100-120 m.a.s.l. and a series of small plateaux and scarps (Simonov, 1977; Smellie and López-Martínez, 2002; Fig. 2). Small lakes occur in the interior of the Meseta, an area which stays almost completely snow free in late summer, except for a few perennial snowpatches. Vegetation is sparse with rocky outcrops and loose clastic material dominating the landscape. The lower areas are the ones where vegetation cover is more frequent, especially at present or past faunal colonies (Michel, 2011).

The climate is polar oceanic, with average annual air temperature at sea-level of -2 °C, summer temperatures of 3 °C, and winter temperatures in August of -7 °C. Annual precipitation ranges between 350 and 500 mm (Øvstedal and Lewis-Smith, 2001), but data from Bellingshausen station show records of 700 mm, part of it during the summer as rainfall events.

The Meseta Norte was selected as a site representative of Maritime Antarctic conditions due to its climate, lack of vegetation and fast snow melt rates during the summer, with frequent late lying snow patches. The area also shows morphological diversity allowing to better assess the spatial variability of the backscatter signal across a variety of slope angles and aspects. The presence of lakes and snow free clast-covered surfaces allowed both for improving geocoding of the TerraSAR-X scenes and ground.

## 3 Methodology

The methodological framework followed in the paper is shown in figure 3 and consisted of: i. A detailed field survey of snow cover characteristics (ground truthing), ii. SAR imagery analysis (remote sensing), and iii. Evaluation of classification methods.

### 3.1 Field characterisation of the snow cover

In order to obtain high quality ground-truthing data, in January 2012, a field campaign was conducted in Fildes Peninsula aiming at characterizing the snow cover at the time of remote sensing imagery acquisition. The area of the Meseta Norte was selected accounting for its variable topography and to the facilitated access from the Chilean Antarctic Station Prof Julio Escudero. Twelve snowpatches with varied slope angle and aspect were mapped and snow pits were dug to describe snow characteristics (Fig. 4).

Snow pits were dug either down to bedrock, or to depths where thick (> 3-5 cm) and difficult to penetrate ice layers occurred. Focus was on the upper 25 cm of the snow pack due to its sensitivity to the propagation of the Xband radar signal. Each of the snow pits was described for snow stratigraphy, grain size and shape and snow density. Due to a failure in the thermometer, no temperature depth profiles were measured. Pervasive moisture in all snow pits showed that snow was wet. As a workaround, ibutton DS1922L single-channel temperature miniloggers were installed at shallow depth (ci. 5 cm) near each snowpit, inside





50 mm cylindrical white plastic photographic film cases. Snow temperature was recorded at 1-hour intervals during a period of several days, which included the dates of satellite imagery acquisition. Fast snow melt and the high infrared absorption of the cases, induced extraordinary diurnal heating inside the cases and daily maximum temperatures were abnormal and thus could not be used to accurately describe snow temperature. Surface melting also induced surfacing of the miniloggers, which had to be reinserted in the morning into the snowpack.

In order to improve geocoding and for a better analysis of the radar imagery, the boundaries of several snowpatches were mapped using a Leica Viva DGPS in RTK mode with a local base station and a rover, allowing for an accuracy of ci. 2 cm for each GPS point. Lake boundaries for improving georeferencing of the satellite images and ground-truthing of water surfaces, as well as bare soil areas (mainly frost shattered debris) were also mapped with DGPS.

## 3.2 SAR imagery classification

Three TerraSAR-X SpotLight SSC (Single Look Slant Range Complex) mode scenes were acquired. SSC products offer a single look of the focused Radar signal, with a scene size of 10 x 5 km. The requested bandwidth was 300 MHz (experimental mode) offering a range spacing of 0.455 m and azimuth spacing of 0.855 m. Table 1 includes additional parameters of the images.

Since the goal was to evaluate the discriminating potential of TerraSAR-X for very high resolution mapping of snow cover, two scenes were acquired for the summer season in 12 and 13 January 2012, the former in HH and the later in VV polarization (Table 1). These dates coincided with the ground-truthing campaign. An additional early spring scene (28 September 2012, HH polarization) was used in order to assess the backscattering for dry snow conditions. The results presented in this paper respect the area of the Meseta Norte, corresponding to the field validation area, which is a subsector of the larger TerraSAR-X scene.

The typical speckle noise (salt and pepper) present in Radar images due to the constructive and destructive electromagnetic interference associated to the scatter implies choosing an adequate filtering phase to each specific area, compensating the noise or emphasizing textures. In this case, Lee filtering with a 9 x 9 window was used to improve the contrast. Calibration to sigma, beta or gamma nought in the radiometric correction phase transfers the digital level captured by the sensor to the Radar backscattering signal (dB) of the reflecting surface (Valenti et al 2008; Small 2011), allowing to compare images acquired in different time lapses and modes. In this study all the calculations were performed with sigma nougth calibration.

Geocoding and terrain correction of the scenes were performed using ranging Doppler analysis and an external digital elevation model of 5 m resolution derived from the Chilean Antarctic Institute topographical map of Fildes Peninsula. This procedure detects geometrical deformations of the original scene due to the off-nadir swapping. Geometric distortions such as lay-over and shadowing are therefore compensated in the final product. Image processing was conducted using the ESA SNAP 3.0 software.



### 3.3 Production and validation of snow cover maps

In order to assess the discriminating potential of TerraSAR-X imagery for high resolution mapping of wet snow distribution, we have first characterized the backscattering signal of the three selected scenes in the ground-truthing areas. For this purpose, the field mapped boundaries of the snow patches, lakes and bare soil were integrated in a GIS, first as a point layer and then

transformed into polygons. Backscattering at the ground-truthing areas was retrieved for the three scenes and analysed in order to identify differences in polarization modes and respective potential for the three surface types. Statistical analyses were conducted to detect backscattering similarities between snow patches and to compare with their topographical setting and snow characteristics obtained from the pits.

The previous approach allowed the identification of backscattering thresholds for the different surfaces and this data together

with ground-truthing was used to test three mapping approaches: i. simple threshold-based surface classification mapping, ii. ratio-based mapping using a dry snow and wet snow scene, and iii. object-oriented mapping. The results of the three classifications are then evaluated by comparison with the reference data.

## 4 Results

### 4.1 Snow characteristics

Twelve snowpits were dug in different snow patches to depths between 45 and 70 cm, at altitudes from 86 to 117 m, at different aspects, located within an area of ci. 0.6 km$^2$ (Fig. 4). Snow surface was generally wet and the downslope concave sectors of several snowpatches showed ponding and slush with pervasive percolation. Snow patch sizes were from tens to hundreds meters wide and slope angle in the sample patches varied from 6 to 34°. Typically, snow pits showed rounded snow grains with melting and frequently clusters of rounded grains (Fig. 5). Grain-size was typically 2 to 4 mm, with grain clusters reaching

40 mm. Compact horizontal ice layers from 2 to 5 cm thick were present in most snow pits. South and southeasterly snow pits showed thicker snow layers and ice layers were generally absent. North and northwest facing snowpatches showed more ice layers, with up to 5 in snow pit (SP) nr. 1. Vertical and horizontal discontinuous refreezing structures revealing percolation in the snow pack were found especially in NW to SE snow patches, normally below 50 cm depth. Snow density ranged from 470 to 600 kgm$^{-3}$ and aspects east to south showed in average denser snow packs (Fig. 6).

### 4.2 Snow patch temperatures

Snow patch subsurface temperatures at 5 cm depth in 12 and 13 January 2012 showed *ci.* 0 °C at all monitored sites between about 23:00 and 7:00 GMT (Fig. 7), corresponding to the period when the sun was very low or below the horizon. As explained in the methodology the packaging of the logger induced extreme heating during the day, with temperatures reaching +16 °C inside some cases. Differences in maxima between snow patches reflected aspect, but diurnal regimes were similar, showing

that there was clear homogeneous signal, reproducing also variability in cloudiness, with synchronous local daytime minima





at all sites. Despite the heating problems, the data confirms that during the days of satellite imagery acquisition, the snow conditions were always close to 0 ℃ at all times and that, even during the night, cooling was small, a fact probably related to the high snow wetness already detected in the snow pits. Such conditions are typical of the Maritime Antarctic, where the daily air temperature range is very small, supporting limited refreezing conditions during the night.

### 4.3 Surface backscattering

The three TerraSAR-X scenes (HH - summer, VV - summer, HH - early spring) show different discriminating potential for snow cover when compared to bare ground and open water surfaces (Fig. 8).

The summer HH-polarization scene showed the best separability between the 3 cover types, with statistically significant differences and clear curves peaking at different backscattering values. Snow peaks between bare soil and water with average -18.0 dB (standard deviation - STD = 2.5 dB). Bare soils show an average of -11.9 dB (STD = 2.0 dB) and water an average of -21.1 dB (STD = 1.6 dB) (Table 1).

The summer VV-Polarization scene shows no discriminating capacity, with a complete mixture between snow and water, with average values of respectively, -11.6 and -13.7 dB. Soil shows higher average backscattering (-6.4 dB), but still overlapping with the former, and especially with snow.

The early spring scene was selected for analysis due to the full snow cover conditions and for preceding the onset of snow melt conditions, therefore representing dry snow. The backscattering values show a complete overlap of the signal of the 3 types of sampled areas, with average values from -5.0 to -6.2 dB and standard deviations of 1.7 to 2.6 dB). This confirms that the sites are covered by a similar type of snow. Secondary peaks in the soil sample areas at -2.4 and -0.7 dB suggest snow free surfaces.

### 4.4 Wet snow patch backscattering characteristics

The HH polarization scene of 12 January 2012 provided the best results for snow patch discrimination and therefore was selected for further analysis and classification. The backscattering for individual snow patches shows significant differences, with average dB ranging from -9.0 in SP2, to -19.8 in SP13 (Table 2). Extreme values range from -4.4 dB in SP8 to -23.6 dB in SP13, with standard deviations from 1.0 dB in SP6 and 13 to 2.8 dB in SP4. These simple statistics show that despite the quasi unimodal distribution (a small bump is visible at around -8 dB but may be an artifact) when considering all snow patches (Fig. 8), very significant differences are found between them.

ANOVA analysis and Kruskal-Wallis non-parametric tests were conducted to identify groups of equivalent response to backscattering. An evaluation of correlation (p-values 95%) by multiple contrast range resulted in three groups (Table 3): i. SP3 and SP12 (underlined), ii. SP5 and SP7 (bold), and iii. the rest, corresponding to a heterogeneous group with no inter-correlation (Fisher´s - LSD - and Bonferroni´s Significative Differences). Figure 9 shows the snow patches ranked by



increasing average backscattering and evidences their moderate to high dispersion when analysing the mean, median and standard deviation. In general, most snow patches show a similar behaviour, with SP2 being the main outlier.

Figure 10a shows the lack of correlation between snow density and backscattering, both in the HH and VV scenes from January 2012. Figure 9b shows that at HH polarization a weak positive correlation exists between backscattering and snow grain-size,

especially evident if removing snow patch nr 2 that shows an anomalously high mean dB of -9.0.

## 5. Image classification

In order to evaluate the applicability of single band HH-polarized data for wet-snow mapping we have tested several methods. An expert-based visual inspection of the HH backscattering sigma-nought scene allows a relatively easy delimitation of wet-snow patches by comparing greyscale values with feature shapes in the terrain evidencing the potential of the scene (Fig. 11).

The purpose of this section is to confront pixel- and object-based classifiers, comparing their performances and potentialities. For extracting the feature values and training the classifiers, we have used a sample set of half of the ground-truthing polygon boundaries obtained in the field survey for snow patches, lakes and bare-soil and the other half to make the validation of the classification

### 5.1 Classification using backscattering thresholds

Given the best quality of the HH-polarisation scene of 12 January, when compared to the VV-scene of 13 January, the former was used for assessing the application of backscattering thresholds and band maths. This is simplest way to evaluate the applicability of the single polarisation backscattering signal to map wet snow. Since data has shown a significant overlap between the three classes, but especially between wet snow and water (see Fig. 8), no accurate threshold is possible to identify between both. However, once the objective is differentiating wet snow from bare soil surfaces, a valid option is using the

threshold between these two classes, which may be defined using the average ± standard deviation for each class: wet snow from -20.5 to -15.5 dB and bare soil from -13.9 to -9.9 dB, while values below -20.5 dB are classified as water. The results of this classification were identified as Thresholds A. Since -20.5 dB is a frequent value for water at the ground truth sites, a second classification was assessed moving the threshold to -19.5 dB, which is mid-distance from the average of wet snow (thresholds b). In order to be able to properly evaluate the performance of the modelling, a random selection of 50% of the

ground truth pixels was used to derive the backscattering thresholds and drive the classification and the other 50% of pixels was used to compute the confusion matrix (Table 4). The large number of pixels in the ground-truthing produced very homogeneous statistical results among the two sets, with maximum differences below 0.03 dB.

Both classifications provide a very good general assessment of wet snow cover distribution, but with the typical noise of pixel-based classifiers (Fig. 10). Most of the wet snow is classified correctly, or as water, but numerous pixels wrongly classified as

snow display in clumped patterns forming fuzzy clusters, with shapes not so clearly defined as the observed snow patches. The confusion matrix shows an overall accuracy of 81.0% for Thresholds A, and 81.1% for Thresholds B.



## 5.2 Classification using simple band-math

Differences in backscattering when comparing a snow scene with a snow-free scene, or with a dry snow scene, have been used by several authors in order to classify ground conditions (Rott and Nagler, 1994; Nagler and Rott, 2000; Malnes et al., 2014). For this purpose, the HH-polarization scene from 28 September 2012 showing a fully snow covered terrain and dry snow, was used as reference, while the HH-polarization scene from 12 January 2012 was the target scene for the classification. Figure 13 shows that from the 3 studied surface types, bare soil is clearly differentiated from lakes, but wet-snow shows a significant overlapping with both lakes and bare-soils, which shows the same limitations as the threshold methods evaluated in a).

Another approach is the classification using a band ratio between the dry-snow scene and the wet-snow scene, aiming at detecting thresholds between surface classes (Nagler and Rott, 2000; Valenti et al., 2008). However, the ratio between the HH-Pol September and the HH-Pol January scenes show very poor discriminating potential with significant mixing between the 3 surface types (Fig. 13). Given the poor results, no image classification for evaluation was attempted using simple band maths.

## 5.3 Classification using an object-oriented algorithm

The pixel-based classifications produced limited results, with the threshold-based maps showing two different patterns of pixels classified as wet-snow: homogeneous patches, with clearly defined limits, which mostly coincided with the ground-truthing snow patches, and small diffuse clusters of pixels, wrongly classified as wet-snow. Given the differences in spatial patterns and the relatively straightforward manual delimitation of the wet-snow by visual inspection on the imagery, new classification tests were conducted using an algorithm which is object-oriented and constituted by 3 main processing steps: filtering, segmentation and classification.

The filtering intends to attenuate the speckle of the radar image in order to enhance the spatial coherence of the image texture or of the structures of the surface. To achieve this goal, a series of mathematical morphology based filters were tested (Soille, 2004). These are region-based filters, which rely on the reconstruction of a 'classic' filter, i.e., by opening or closing (Salembier and Wilkinson, 2009). The filter that performed better is based on the removal of the image extrema with a contrast criterion, that is, on suppressing all maxima and minima whose height/depth are lower than a given threshold level $h$ (Soille, 2004).: this value was fixed at 10% of the backscattering range of variation on the whole HH scene. The output of this filter is shown in figure 14-a, where the structures of the landscape are now more evident than in the initial image.

The segmentation consists of delineating the homogeneous regions, often referred as objects (Blaschke, 2010), of the filtered image. The underlying idea is to classify later the basic elements of the texture (the objects) instead of the basic elements of the digital image (the pixels), since the availability of additional descriptors of the image can greatly improve the decision performance. The segmentation is based on the watershed transform (Soille, 2004), followed by a post-processing task to merge similar adjacent regions. The final watershed lines corresponding to the segmentation of the filtered image are shown superimposed to it in figure 14-b.



Finally, in the third step, the classification of the segmented objects is performed. It is a supervised classification approach, meaning that typical features of the objects are used to train a classifier. In the current situation, the classifier that achieved better results is Support Vector Machine (SVM). SVM is a supervised kernel method (Vapnik, 1995) that uses an implicit transformation to a higher dimensional space in order to achieve a good separability by means of a linear classifier. It also has

the ability to handle data with unknown statistical distributions using small training sets. The classification of the segmented objects is based on a set of intensity, geometric and textural descriptors of each object. The SVM kernel selected is the RBF-Radial Basis Function with the parameters gamma = 0.03 and C = 1000. The classified image of the study site is shown in figure 15. The good visual agreement between the classified image and the input radar image indicates already how well the classifier performed.

The confusion-matrix shows the good quality of the classification, with an overall accuracy of 92% and Kappa equal to 0.88 (Table 3) (Congalton, 1991), indicating the adequacy of the proposed method to separate water, snow and soil in radar images of ice-free regions in Maritime Antarctica. The integration into the same processing sequence of some of the most appropriate filters to deal with the spatial arrangement of textures, the a-priory delineation of the objects constituting the landscape and the use of one of the most robust classifiers, are the keys for the performances obtained. The only issues in the classification

arise in snow patches facing west to northwest, where a significant part of the area was classified as bare soil.

## 6 Discussion

Snow characteristics in the Meseta Norte at Fildes Peninsula in 12 and 13 January 2012 have been described by field mapping of test snow patches and by analysing snow pits. Snow distribution showed a typical Maritime Antarctic summer melt pattern with snow patches from tens to hundred meters large concentrating in concavities and prevailing in south facing slopes. The

snow was in advanced melting stage, with isothermal near 0 °C temperatures even during the night, showing the delay effects of latent heat exchange during freezing. The downslope sectors of several snow patches showed ponding and saturated slush. Snow pits down to ci. 70 cm showed that grain-size was generally 2-4 mm and crystals showed melting with frequent clustering evidencing advanced metamorphism and warm conditions (Braun, 2001). Snow pits reveal frequent ice layers associated to melting events, refreezing and new snow accumulation. Such ice layers were more frequent in snow patches facing NW to SE,

while south facing snow patches showed rare ice layers. This pattern reveals the effects of insolation and possibly warm air advection on snow stratigraphy and the significance of snow melt events during the snow accumulation season. Snow densities agree with the relatively mild climate of the region, with high values ranging from 470 to 600 kg/m$^3$, close to the typical late melting season 350 to 550 kg/m$^3$ indicated by Dewalle and Rango (2008).

X-band radar backscatter is essentially influenced by the characteristics of the upper 15 cm of the snow pack (Rees, 2006; Rott

and Nagler, 2013). Near the surface, most of the snow patches showed a lack of ice layers, coarse grained snow and high densities, ranging from 470 to 600 kg/m3. The HH polarization scene showed a better discriminating potential between wet snow, bare soil and water, than the VV-pol scene, which completely merged the water and snow signals, while also showing important overlap with bare soil.





Snow backscattering in the HH-pol scene of 12 January 2012 showed values from -4.4 to -23.6 dB, with a mean of -18.0 and a standard deviation of 2.5. The mean value is in agreement with the wet snow signal from other regions, as described by Shi and Dozier, 1997 (-5 to -20 dB) and in range with the experimental margin described by Ulaby and Stiles (1981). The minimum was measured at SP2, with an average slope angle of 34°, which was probably not well resolved with the DEM.

Given the limitations of the VV-Pol scene, the HH-Pol scene was selected to test the use of single band classification methods for identifying wet snow. The results showed a significant overlap between the signature of wet-snow and lake water. Wet snow showed higher dB and visual inspection shows that spatial distribution of values in snow patches is more uniform, whilst lakes show higher speckle. The mixed signal between wet snow and water generates a large number of errors when conducting a pixel based image classification, with numerous pixels classified as water in slope sites where bare soil occurs and a

significant mix of snow and water. One of the reasons for the poor discrimination potential is the high moisture content of snow, and also the high moisture of soils during the summer, with saturation occurring in many locations. The tests carried out with simple thresholds and band maths did not provide robust results. However, the threshold based maps allow identifying the snow patches, although with a significant noise and too much snow in bare soil areas.

Supported by the visual inspection of the HH-Polarization scene and with the terrain-based expert knowledge suggesting that

snow patch boundaries could be easily identified in the scene, an object-oriented approach was tested as an alternative for the limited performing pixel-based methods. Filtering and image segmentation were important steps for cleaning the noisy areas and the classification results improved very significantly, with an overall accuracy of 92%. The resulting classification was very good. Noisy areas were removed and a very good overall performance was obtained. Problems still occur in snow patches facing northwest, which have been misclassified as bare soil. This problem has also been found when analysing the VV-

polarisation scene and is probably related to artefacts associated with the geometry of acquisition, which was along an ascending orbit, right looking.

## 7 Conclusions

TerraSAR-X imagery shows clear advantages in high cloudiness environments when compared to optical images, since the radar signal traverses the cloud cover and is not dependent on daylight. But the Radar signal structure is very dependent of the

topography and the dielectric variables of the terrain, and in the case of the snow, on grain size and snow water equivalent, implying a large variability in backscattering according to local factors and time. The acquisition mode is very relevant to achieve an adequate spatial resolution with minimal geometric distortions. The use of Single Look Slant Range Complex (SSC) images permitted a sophisticated terrain correction, excluding lay-over and shadowing effects with a precise external DEM. The High Resolution Spotlight Mode and a refined speckle filtering that can be useful determining the limits of the snow cover

to adjust much better with the in terrain data collected.

In the present study we conducted a very detailed survey of snow conditions in two days in the austral summer of 2012, with simultaneous acquisition of two TerraSAR-X scenes spotlight mode in HH and VV polarization modes, and a third HH polarization scene was obtained in 28 September 2012 as a reference dry snow scene. Snow patches were in advanced melting





stage, with wet and coarse-grained snow at all studied sites and ponding in the downslope sectors of some snowpatches. As a consequence of snow melt and also of active layer thaw, the bare soils of Fildes Peninsula showed significant moisture content. The analysis of the TerraSAR-X scenes and the comparison with ground-truthing from snow patches, lakes and bare soil test areas showed that the only scene with potential for discrimination of the three surface classes was the one obtained with HH-

Polarization. However, despite different average backscattering, still significant mixture occurred between the 3 classes. With the objective of mapping wet-snow distribution, we have tested single band pixel-based classification methods and an object-oriented approach. After several tests with the latter, this has proven to be the one providing best classification results, with overall accuracies of 92%. Some inaccurate classifications were obtained in northwest to west facing snowpatches, and especially in steeper slopes. The reason for this is probably associated with the geometry of image acquisition and further

research is needed to mitigate this issue.

The method presented here using spotlight mode imagery together with detailed synchronous reference data offers for the first time a very high resolution mapping of snow patches in the Maritime Antarctic, allowing identifying features with a scale of a few meters. Given the lack of knowledge on snow melt in the ice-free terrains of the Antarctic Peninsula, the present results show that X-band imagery can be used as a good approach for monitoring snow melt patterns during the summer in key areas.

Such an approach is especially useful for monitoring ecosystem dynamics (i.e. at GTN-P, CALM-S or LTER observatories), modelling permafrost and active layer thaw, but also for remotely assessing snow conditions before opening summer research stations and thus implementing better planning for deploying equipment and personnel.

**Acknowledgements**

This research has been funded by the Portuguese Polar Programme and the Fundação para a Ciência e a Tecnologia under the

projects SNOWCHANGE and PERMANTAR-3 (PTDC/AAG-GLO/3908/2012). Imagery was obtained through the DLR TerraSAR-X project LAN1276. The authors warmly thank the Instituto Antártico Chileno for the logistical support provided at Prof. Julio Escudero Research Station in Fildes Peninsula.

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





| Date | Time (UTC) | Orbit cycle | Pass | Incident angle (º) | Polarization |
|---|---|---|---|---|---|
| 12/01/2012 | 23:32:59 | 153 | Ascending | 45.626 | HH |
| 13/01/2012 | 23:15:59 | 153 | Ascending | 29.875 | VV |
| 28/09/2012 | 8:39:59 | 176 | Descending | 25.259 | HH |

**Table 1: Characteristics of the TerraSAR-X scenes used for snow mapping in Fildes Peninsula.**

| | Bare soil (dB/STD) | Snow (dB/STD) | Water (dB/STD) |
|---|---|---|---|
| HH - summer | -11.9 / 2.0 | -18.0 / 2.5 | -21.1 /1.6 |
| VV - summer | -6.4 / 2.0 | -11.6 / 3.1 | -13.7 / 3.3 |
| HH - spring | -5.0 / 2.4 | -5.7 / 2.6 | -6.2 / 1.7 |

**Table 2: Backscattering characteristics of the sampled snow patches, lakes and bare soil areas. STD – Standard Deviation.**

| Snow patch | Backscattering (dB) | | | | | | SWE (cm) | Surface (0-5 cm) | | | Subsurface (5-10 cm) | | |
|---|---|---|---|---|---|---|---|---|---|---|---|---|---|
| | Mean | std dev | Max | Min | Slope (°) | Aspect | | Grain Size (mm) | Density (kg/m3) | Ice layer | Grain Size (mm) | Density (kg/m3) | Ice layer |
| 3 | -18.2 | 1.3 | -14.1 | -22.0 | -18.1 | 15 | N360 | 4.7 | 3 | 472 | no | 2 | 487 |
| 12 | -18.2 | 1.8 | -8.2 | -22.7 | -18.5 | 9 | N130 | 5.5 | 4 | 550 | no | | 519 |
| 5 | -18.8 | 1.1 | -14.9 | -23.0 | -18.8 | 20 | N10 | 4.7 | 3 | 472 | no | 4 | 487 |
| 7 | -18.8 | 1.4 | -14.8 | -23.5 | -18.7 | 8 | N290 | 4.6 | 3 | 456 | no | 2 | 487 |
| 1 | -15.0 | 1.8 | -9.2 | -19.6 | -15.0 | 11 | N270 | 4.7 | 3 | 472 | no | 4 | 487 |
| 2 | -9.0 | 1.4 | -4.5 | -14.9 | -8.7 | 34 | N270 | 5.0 | 2 | 503 | no | 4 | 550 |
| 4 | -16.2 | 2.8 | -8.0 | -20.7 | -17.0 | 20 | N180 | 4.7 | 4 | 472 | yes | 3 | 519 |
| 6 | -19.6 | 1.0 | -13.6 | -22.8 | -19.6 | 9 | N130 | 4.7 | 2 | 472 | yes | 1 | 550 |
| 8 | -17.8 | 1.8 | -4.4 | -21.6 | -18.0 | 6 | N170 | 5.2 | 2 | 519 | no | | 519 |
| 10 | -18.5 | 1.6 | -13.2 | -23.4 | -18.5 | 8 | N70 | 6.0 | 3 | 597 | no | | 597 |
| 11 | -17.2 | 1.6 | -13.1 | -21.4 | -17.1 | 10 | N280 | 5.2 | 3 | 519 | yes | 3 | 519 |
| 13 | -19.8 | 1.0 | -16.6 | -23.6 | -19.7 | 20 | N120 | 5.2 | 2 | 519 | yes | | 519 |

**Table 3: Snow patch characteristics and backscattering in HH-Polarization (12 January 2012).**




| | Thresholds A | | Thresholds B | | Object-oriented | |
|---|---|---|---|---|---|---|
| | Prod. Acc. (%) | User Acc. (%) | Prod. Acc. (%) | User Acc. (%) | Prod. Acc. (%) | User Acc. (%) |
| Wet snow | 85.2 | 72.1 | 66.8 | 82.3 | 87.49 | 95.11 |
| Water | 73.7 | 91.3 | 88.8 | 80.2 | 95.16 | 100.0 |
| Bare soil | 96.3 | 81.8 | 96.3 | 81.8 | 100.0 | 69.65 |
| Overall accuracy (%) | 81.0 | | 81.1 | | 92.36 | |
| Kappa | 0.69 | | 0.68 | | 0.88 | |

**Table 4: Performances for the 3 tested classifications. Classification Thresholds A: backscattering threshold water – wet snow at -20.5 dB, Classification Thresholds B: backscattering threshold water – wet snow at -19.5 dB, and Object-oriented approach.**



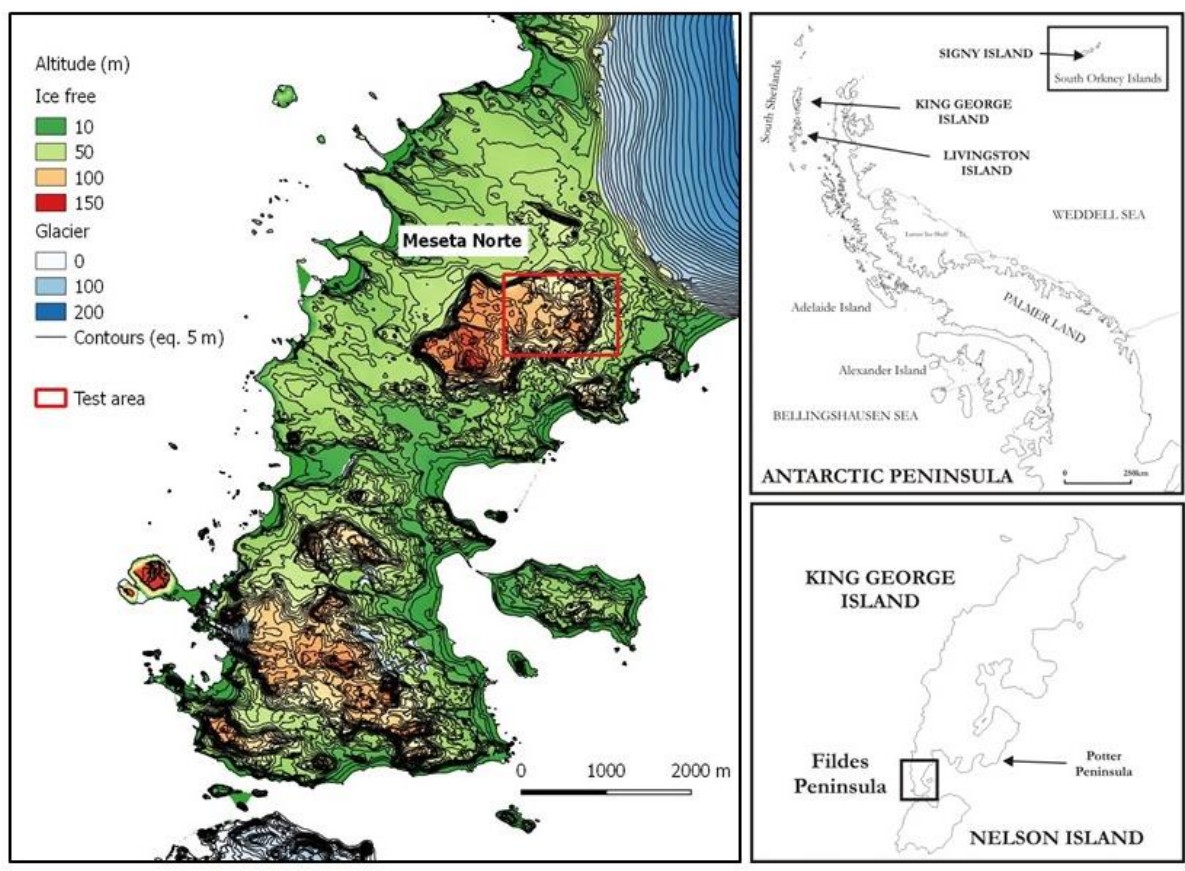

**Figure 1: Location and topography of Fildes Peninsula and the Meseta Norte test site.**



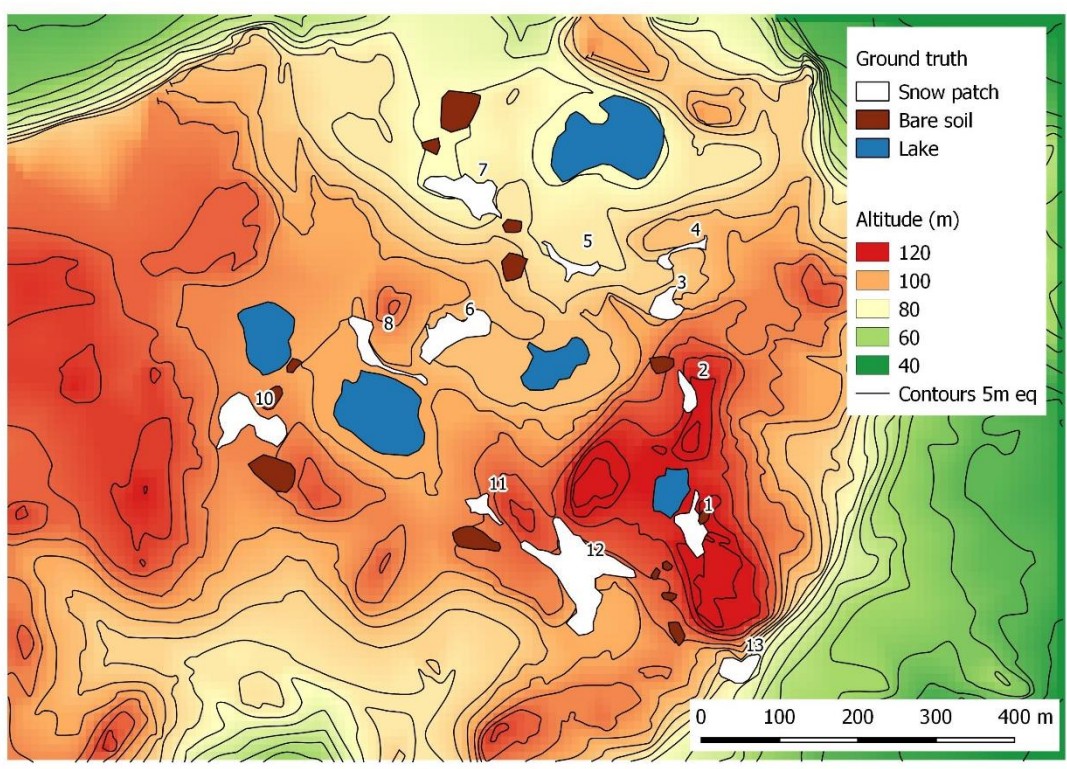

**Figure 2: Topographical setting of the Meseta Norte test area in Fildes Peninsula with the ground truthing mapped in the field. Snow patches are numbered as in the paper.**





**Figure 3: Methodology for the evaluation of the potential of Spotlight mode TerraSAR-X imagery for high resolution snow cover mapping.**



**Figure 4: Overview of the snow conditions in the sampled snow patches during the field survey in the Meseta Norte in January 2012.**





**Figure 5: Snow pit description for the studied snow patches in Fildes Peninsula.**

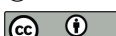


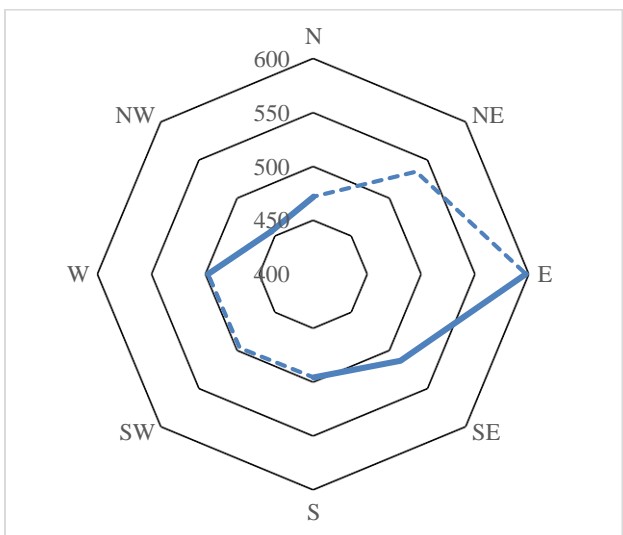

**Figure 6: Surficial snow density (kg/m³) according to aspect in the studied snow patches in Fildes Peninsula. Dashed line represents estimated values at NE and SW calculated by averaging between neighbouring orientations.**

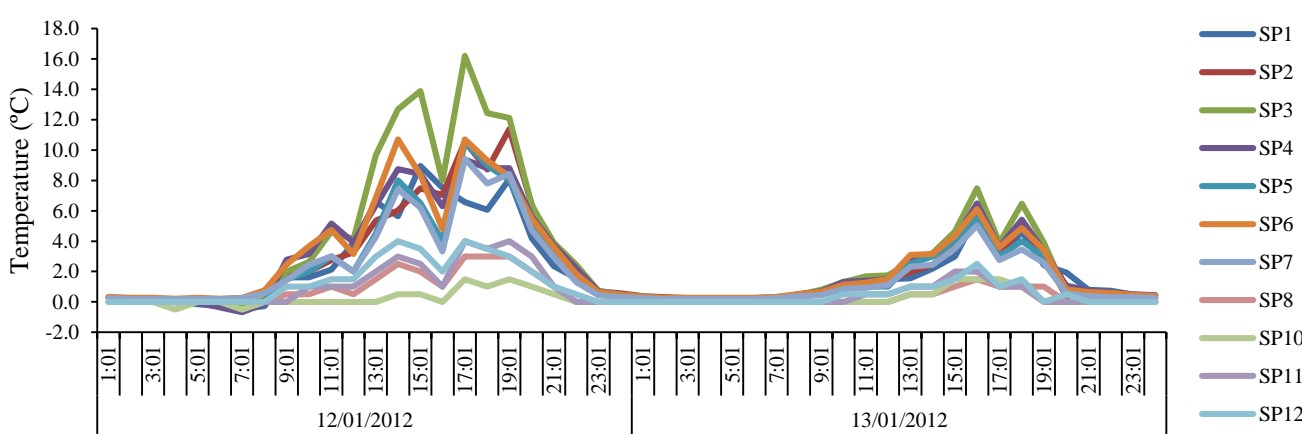

**Figure 7: Snow patch temperatures measured at 5 cm depth from 12 to 13 January 2012. The peaks in the maxima relate to anomalous overheating of the minilogger case. SP1-SP12 are the snowpatch numbers. SP13 was not monitored.**





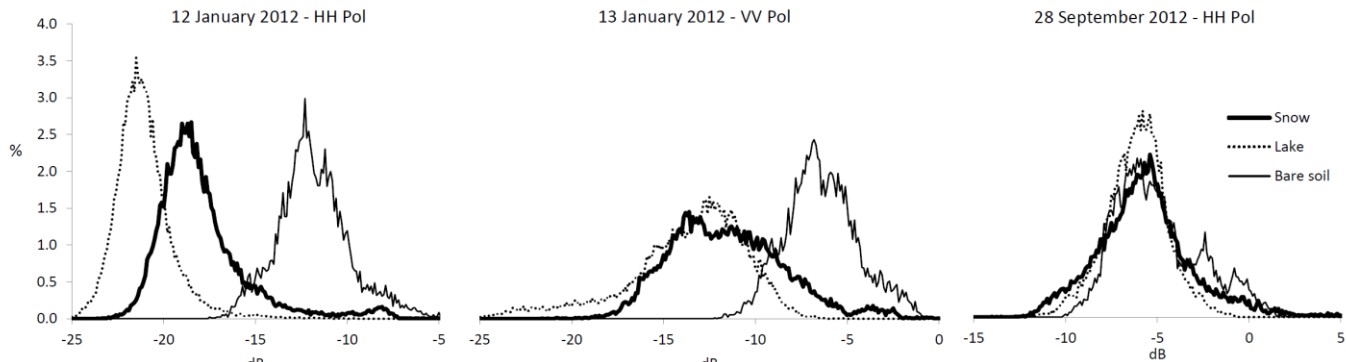

**Figure 8: Radar backscattering of snow, water and soil ground truthing areas for the three selected TerraSAR-X scenes: a. HH-Polarization (12/01/2012), b. VV-Polarization (13/01/2012) and c. HH-Polarization (28/09/2012).**

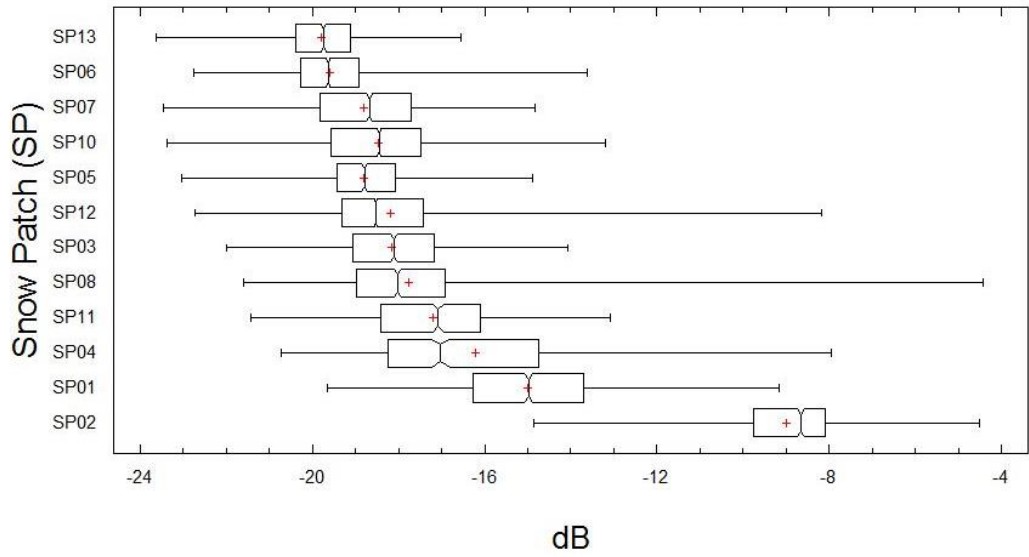

**Figure 9: Box and whisker plots of backscattering of individual snowpatches in the HH polarisation scene from 12 January 2012.**





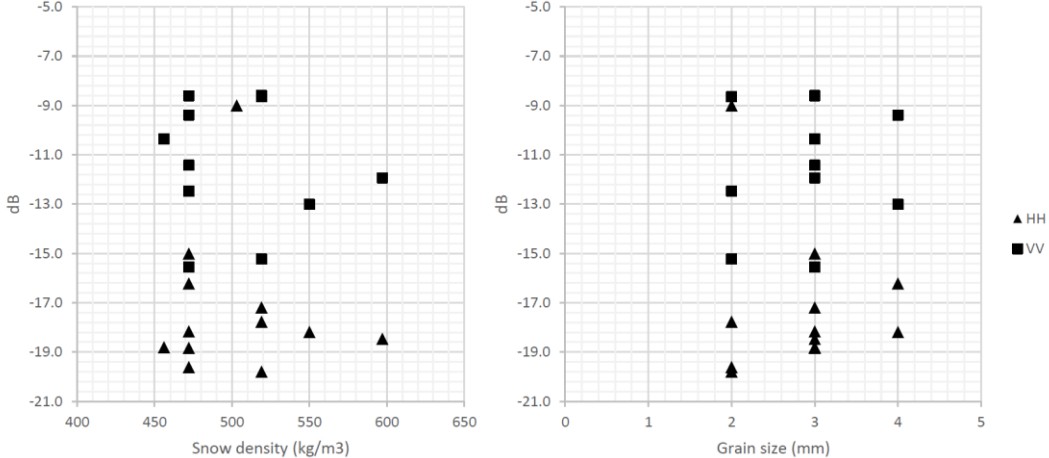

**Figure 10: Scatterplots of snow density (a) and snow grain-size (b) with backscattering for the studied snow patches.**

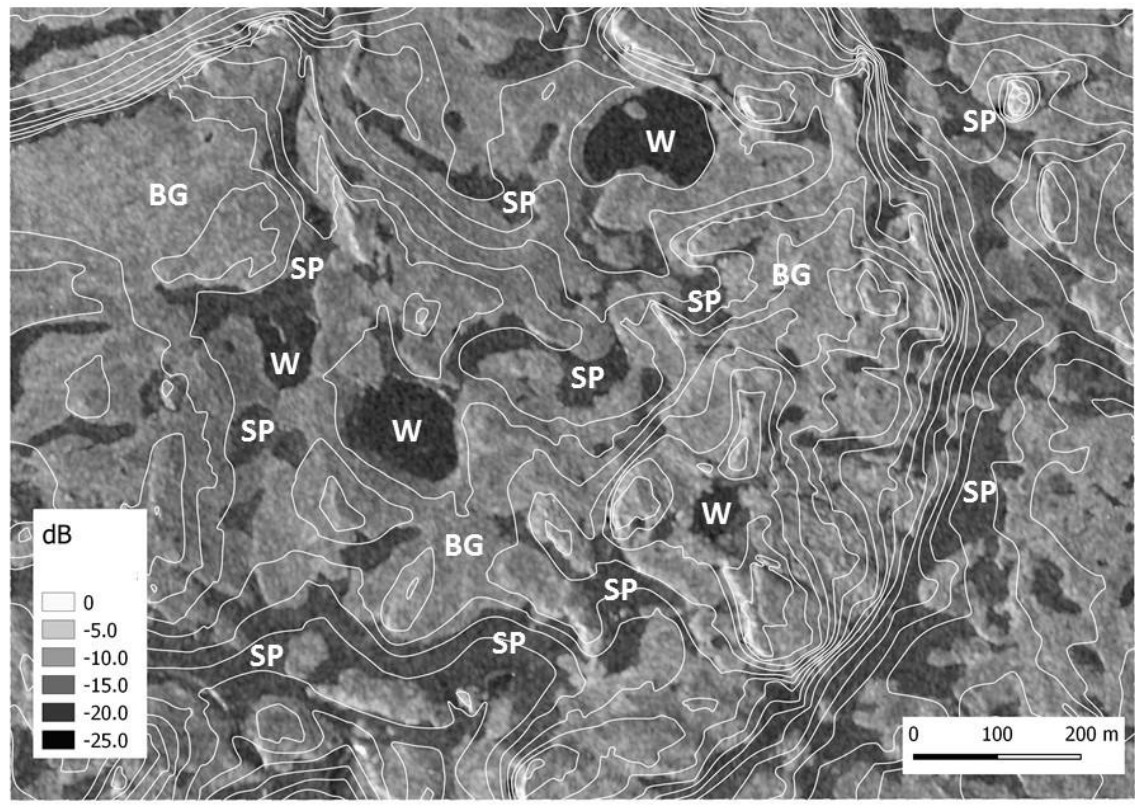

**Figure 11: TerraSAR-X SpotLight mode HH polarization backscattering (sigma-nought) scene from 12 January 2012 with examples of visual interpretation of bare ground (BG), water (W) and snow patches (SP). Countour lines at 5 m equidistance.**



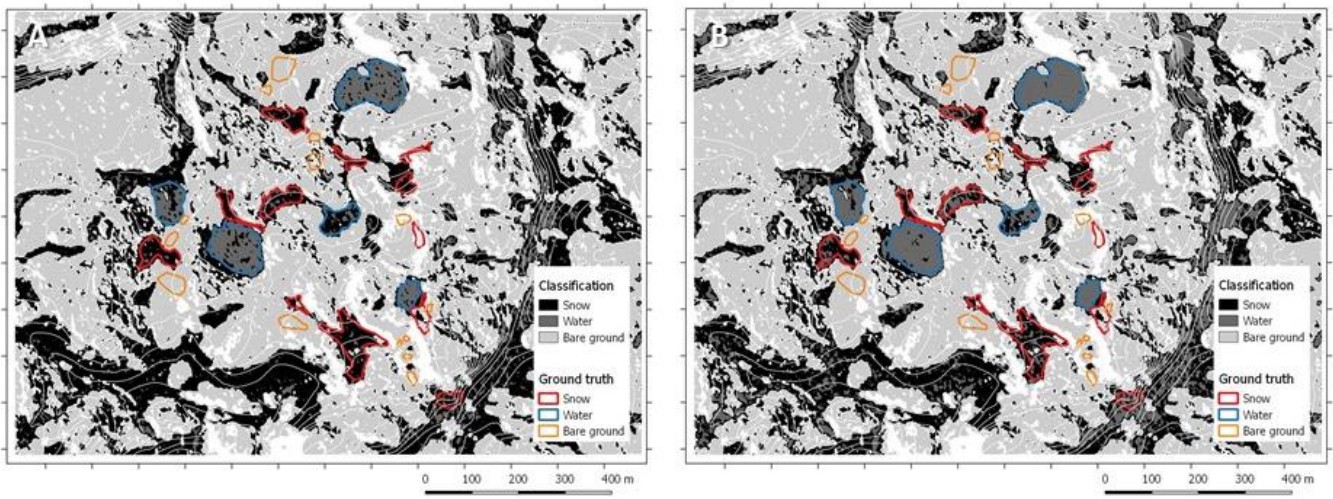

**Figure 12: Classification results using: a. backscattering threshold water – wet snow at -20.5 dB, b. backscattering threshold water – wet snow at -19.5 dB.**

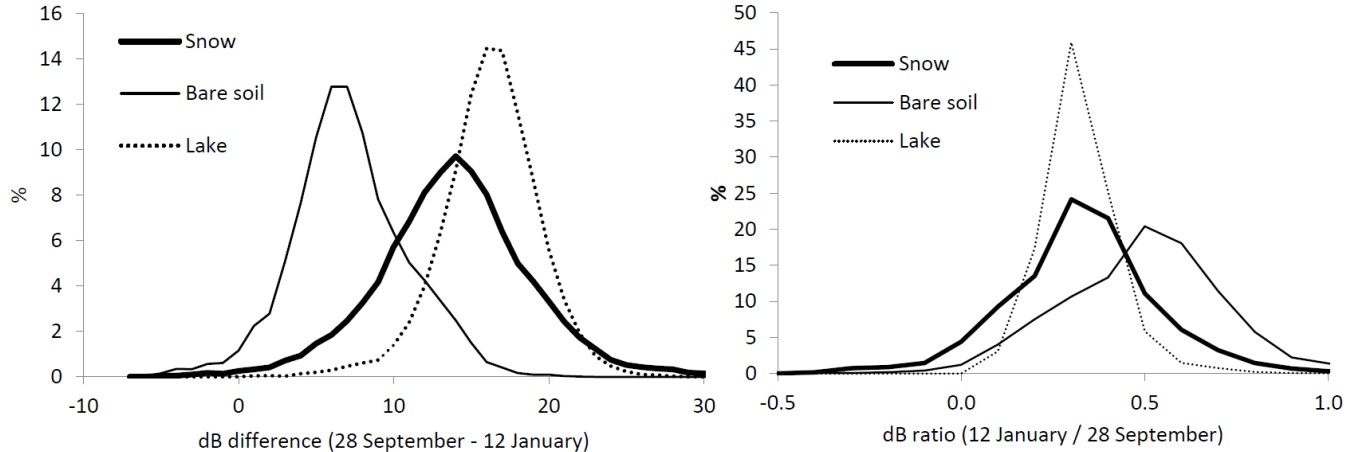

**Figure 13: Discrimination of wet-snow, bare-soil and water using simple band maths: a. difference between the dry snow scene (28 September 2012, HH Pol) and target scene (12 January 2012), b. band ratio between the dry snow scene and the target scene.**





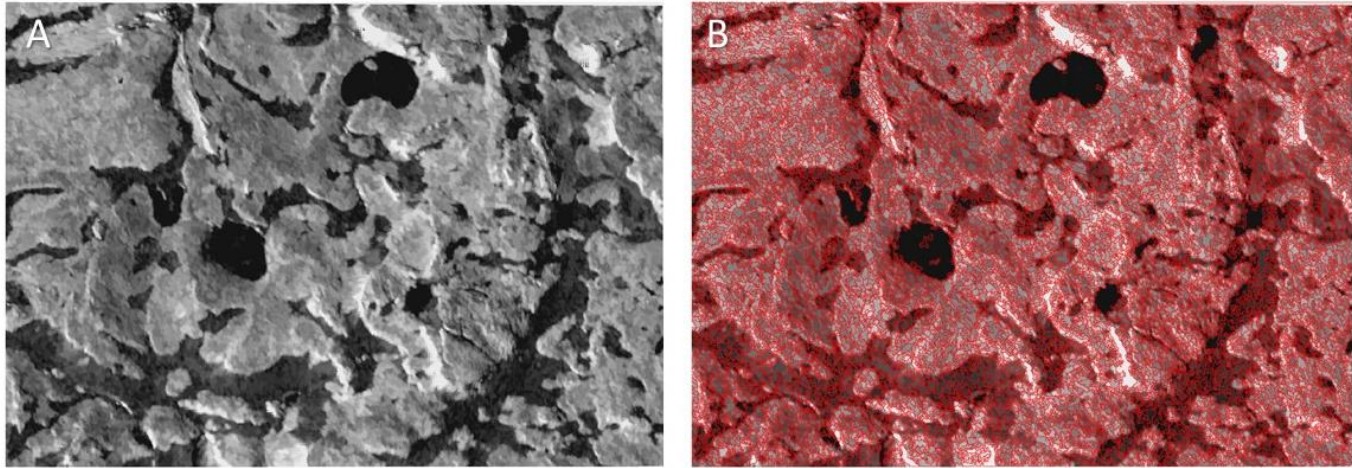

**Figure 14: Initial steps on the object-oriented classification scheme: a. filtered image by suppression of its extremes values with a contrast criterion (mathematical morphology h-max and h-min operators), b. segmentation by watershed with the delineation of the objects of the filtered image.**

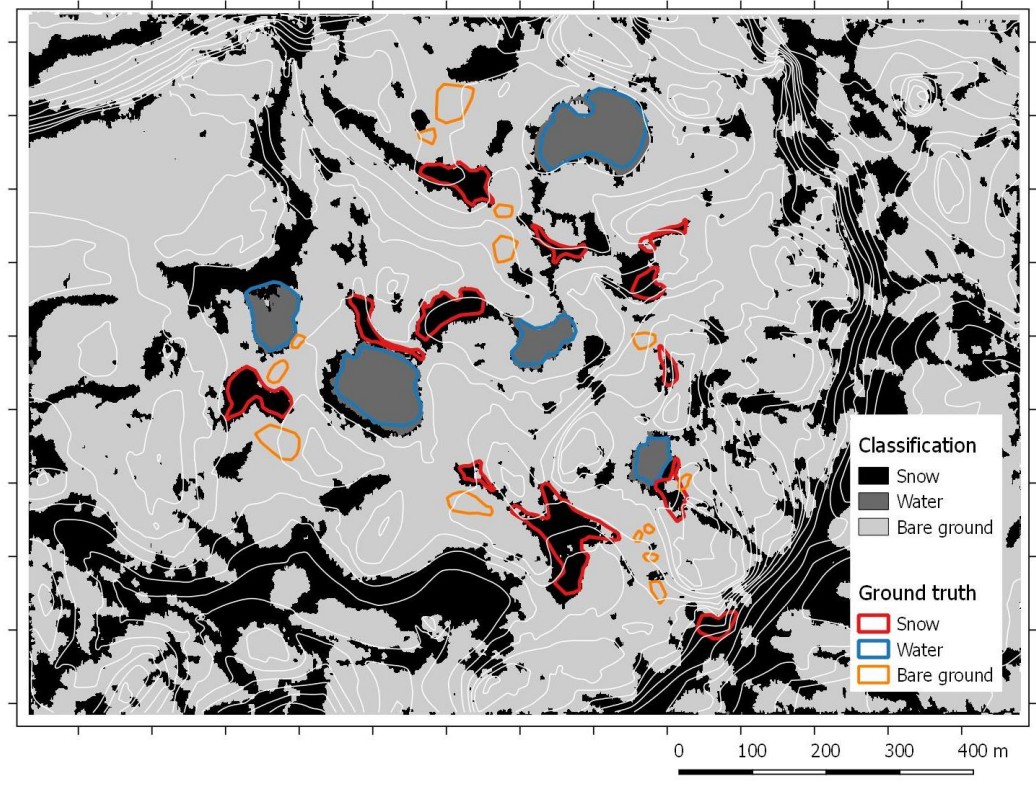

**Figure 15: Distribution of wet-snow in the Meseta Norte using an object-oriented classification with SVM - Support Vector Machine.**