# Peer review of "Evaluation of single-band snow patch mapping using high resolution microwave remote sensing: an application to the Maritime Antarctic"

_The Cryosphere, 2016_

## Referee Comment (RC1) · Anonymous Referee #1 · 28 Sep 2016

**1   General comments:**

The manuscript presents an approach to map snow patches in the Maritime Arctic using high-resolution SAR data. The authors examine different classification approaches and have conducted an extensive field campaign in order to evaluate the obtained results. They find a SVM-based classification approach suitable for mapping wet snow patches within their study region. The paper is clearly structured, the scientific methods are described in detail and the results are well presented. Nevertheless, the following points need to be addressed by the authors.

[Figure]

**2 Specific comments:**

There is an incidence angle dependency of the backscattered signal. Depending on the local incidence angle of your terrain (SAR scene incidence angle + terrain slope), this can become significant. You cite this effect in the introduction, and you also observe it at steeper terrain slopes, where your classification fails. If your method is intended for wider use (and in your abstract, you mention a possible operational application), how can you handle incidence angle dependency?

You found the HH scene to be better suited than the VV scene acquired on the following day. Do you have an explanation for this observation?

The water bodies you observed show very low backscatter. How would your classification approach handle wind-roughened water which can become very bright due to Bragg scattering?

1 Introduction:

In line 24-25, you state that "Most applications have been developed for regional scale mapping, but for higher resolution approaches they lack quality." This is a very strong remark, please elaborate on that.

3.1 Field characterization of the snow cover:

First of all, I think it is a very good idea to comment on the failed temperature measurements and to give a detailed analysis of the possible cause. Still, I am missing a description of the other methods of measurement. How did you measure grain size and how do you define grain size in the first place? How did you measure snow density?

3.2 SAR imagery classification:

This section does not actually describe the classification method, maybe you should rename it to "SAR image processing" or something similar.

4.2 Snow patch temperatures:

Did you consider using external temperature measurements, e.g. from AWS or Reanalysis data, for your study? Given the narrow range of temperatures for your test site, it would have been also interesting to have temperatures available for the September image.

4.4 Wet snow patch backscattering characteristics:

On page 8, lines 4-5, you state that "Figure 9b shows that at HH polarization a weak positive correlation exists...". I cannot see any correlation in the figure and suggest to rephrase this sentence.

5.3 Classification using an object oriented algorithm:

Here, you use a set of morphological filters to suppress speckle and to obtain more homogeneous regions. If the quality of your threshold-based classification suffers from the same noise characteristics, then why didn't you use that set of filters for all classifications?

**3 Tables and Figures:**

- Table 3: There is something seriously wrong with this table. From column 7 on, it does not make any sense.

- Table 4: What do you mena by prod. acc. / user acc.? Please explain the abbreviations.

- Figure 12: The legend is very hard to read, please make it bigger. If you have 4 classes in the image (white, light and dark gray, black), why do you only have 3 of them in the legend?

- Figure 14b: This figure is very hard to interprete, since it looks just like Fig 14a tinted red. Maybe a zoomed-in region could provide a higher level of detail?

**4 Technical corrections:**

- page 3, lines 20-22, "Mapping of the later...": This sentence got a bit lost, it seems.

- page 4, lines 14-16, "... geocoding of the TerraSAR-X scenes and ground.": There is something missing here.

- page 4, line 30, "Pervasive moisture...": This sentence appears to be a bit out of context, maybe shift it up a bit, after "Each of the snow pits...".

- page 8, lines 15-16, "Given the best quality...": This sentence is a bit confusing, please rephrase. The next sentence is missing a "the".

- page 8, line 24, "thresholds b": If you use uppercase on the other scenarios, use it here as well.

- page 8, line 29, Fig. 10: should probably be Fig 12.

- page 10, line 25: "snow patches showed rare ice layers": I suggest rephrasing to "...snow patches rarely showed ice layers"

- page 11, lines 26-27, "The acquisition mode is very relevant...": I don't really understand what you mean to convey with this sentence.

---

## Short Comment (SC1) · 4 Oct 2016

This is a very interesting and useful manuscript that highlights the potential of very high resolution, X-band SAR imagery for large cartographic scale, snow cover mapping. Notwithstanding, I find the manuscript's structure somewhat ineffective, particu- larly with regard to method description. Relatedly, the methodology used to evaluate the performance of the classifiers is not well explained and seems inadequate (does not translate actual performance). I hope that the following comments come across to the authors as constructive in intention; some of the issues I highlight below are issues that I deal with in my own research and have particular interest on.

Section 3.3 is titled "Production and validation of snow cover maps" but does not explain how the validation was done. I think it would be beneficial to separate classi- fication algorithms ("Production of snow cover maps") from validation. Only after the results (section 4), in section 5, is it explained that the ground truth data was divided into classification and validation sets. I would additionally suggest explaining how the separation between classification and validation sets was performed (e.g., random?) as well as adding considerations on the representativeness of the samples (different configurations will significantly affect computed performance). Considering the small size of the reference dataset, for a minimum-bias assessment, the performance of the preferred classifier should be trained and evaluated using multiple training and vali- dation sets (from multiple, different partitions). As is, it would be useful to have the classification and validation polygons discriminated in one or all of the results' maps; or, maybe, just remove the patches used for classifier development from those maps.

The study area is quite small (< 1.5 sq. km?), yet the reference data is significantly spatially restricted; although it could be difficult to analyze snow properties for all snow patches, it is clear that wet-snow conditions are widespread – why only some patches were mapped in the field? Additionally, since the presented method for snow mapping involves classifying non-snow land covers, having more extensively field-mapped the non-snow classes would have enabled a more reliable performance assessment in- dependently of the snow cover mapping. My concern is that the presented values of Kappa, etc., though encouraging, do not properly convey the performance of the clas- sifications. For example, in Fig. 15, in two instances, the snow ground-truth polygons (the northernmost and southernmost polygons) are much smaller than the SAR-image derived snow patches they overlap. Do those (red) polygons represent the actual ex- tent of the snow patches? If so, it means that the overmapping for the snow class is much more significant than the performance measures suggest, and thus actual per- formance is lower than the computed performance; i.e., the geometry and distribution of the ground-truth areas would have been a strong determinant of measured perfor- mance. If not, what was the rationale for mapping only a portion of the snow patch?

It would be more effective to describe the used statistics (evaluation of the different polarizations for land cover class discrimination; comparison of the classification algorithms; automated classification evaluation) under methodology. Currently, they are essentially referred for the first time in or after the results section.

In line with a comment from reviewer #1, section 3.2 deals with data and data (pre)processing, not with image classification as suggested by the respective title; ideally, there would be a correspondence between the 3 items highlighted in the text right after section 3 header (Methodology), and section 3 level-2 headers.

Section 5 is composed of results and thus should be under the results section (section 4). The method descriptions under section 5 would move to the methodology section.

---

## Short Comment (SC2) · 24 Oct 2016

Thank you very much for the questions and remarks that we believe contribute to clarify several parts of the manuscript. Below, we present a detailed answer to the questions you have raised concerning the text and tables, while we have clarified the manuscript accordingly, and have also included your technical corrections. We will send this in the revised manuscript and we hope that you find our answers sactisfactory.

Question : ' There is an incidence angle dependency of the backscattered signal. Depending on the local incidence angle of your terrain (SAR scene incidence angle + terrain slope), this can become significant. You cite this effect in the introduction, and you also observe it at steeper terrain slopes, where your classification fails. If

your method is intended for wider use (and in your abstract, you mention a possible operational application), how can you handle incidence angle dependency? ' Reply : This is a significant issue that will always include limitations linked to the difficulties on accurately modelling the backscattering, but that we think could be mitigated with improved digital elevation models (e.g. better accuracy and better resolution). An approach could be by using UAV-based aerial photo surveying and DSM generation. However, such models will never be perfect due to temporal changes in snow accumulation patterns inducing varying local snow morphologies. Slope have been widely studied to introduce geometric corrections (e.g. Mi et al, 2014; Small et al, 2010), but we have adopted a simpler but robust approach through Range Doppler Terrain Correction, taking into account the advantage of a 5 m DEM. However, as we show, some incidence angle + slope relationships will remain difficult to resolve. In the procedure that we have applied, the incidence angle and terrain slope are both considered in the absolute radiometric calibration to sigma nougth in ESA-SNAP software (Kellndorfer et al, 1998), and in the subsequent phase of Range Doppler Terrain Correction. Using imagery with showing multiple SAR incident angle backscattering responses would be the best approach to infer a more complete radiometric perfomance of the terrain signal and is a recommended practice to regionalise the results. Unfortunately, in this case only two scenes were available (HH with 45.626 incident angle and VV with 29.875 incident angle). The original plan was to have more imagery, but not all acquisitions were guaranteed. We will improve the discussion of this issue in the revised version of the manuscript.

Question: You found the HH scene to be better suited than the VV scene acquired on the following day. Do you have an explanation for this observation? Reply : We have found a similar behaviour when using Envisat ASAR imagery for Deception Island (Mora et al 2013), as well as other authors, such as Baghdadi et al., 1998 using polarimetric SAR data at C-band for the classification of land covers (open areas, lakes ice, and forests, all covered with wet snow) and they have also concluded that HH-polarisation is better than VV-polarisation. The backscattering behavior is dependent
on the dryness of the snow, on the incident angle and on the roughness of the surface. For classification purposes the most important issue is the separability between classes and in this case, it seems that HH is more appropriate to separate between water, bare soil and wet snow. Additionally, VV polarisation is more sensitive to water roughness changes. In the case of our scenes, the HH scene shows a higher incidence angle, which improves resolution in a terrain with an irregular topography (Woodhouse, I., 2006), such as the study area. We will improve the manuscript, by adding this discussion.

Question : The water bodies you observed show very low backscatter. How would your classifi- cation approach handle wind-roughened water which can become very bright due to Bragg scattering? Reply : This is a very good question. In order to implement our approach operationally, the lake surfaces should be masked after an initial detection. This would pose issues where lake water levels vary very significantly, or where lakes cover a large percent of the terrain, but neither is the case in the ice-free areas of the Maritime Antarctic. So, an initial assessment of lake boundaries, either using imagery in low wind conditions, or using optical imagery, could be used to create a lake mask. We will introduce this discussion in the manuscript.

Question : In line 24-25, you state that 'Most applications have been developed for regional scale mapping, but for higher resolution approaches they lack quality.' This is a very strong remark, please elaborate on that. Reply : You are right. We will clarify the sentence and delete the last part. Essentially, the literature lacks published results on the use of SAR for snow mapping and very high resolution (metric).

Question : 3.1 Field characterization of the snow cover: First of all, I think it is a very good idea to comment on the failed temperature measurements and to give a detailed analysis of the possible cause. Still, I am missing a description of the other methods of measurement. How did you measure grain size and how do you define grain size in the first place? How did you measure snow density? Reply : You are right. We will improve the description of the snow-pit characterization. Grain-size was measured by

carefully collecting small amounts of snow from each of the layers of the snow pack and by depositing them in a black tissue for contrast. They were then observed with a 10x magnifier, which allowing for measuring and describing the grain shape and size. Grain size (or crystal size) showed variability some within each layer and our descrition encompasses the mean grain-sizes, but when variability was large, we included the more frequent dimensions (i.e. 1-2 mm). Snow density was measured by carefully collecting snow from each snow layer without disturbing the density, using a metal box with a volume of 212 cm3. From each layer, 3 boxes were collected, adding up 636 cm3, which were inserted in a plastic bag and weighted using a digital spring scale, and mass converted to density.

Question : 3.2 SAR imagery classification: This section does not actually describe the classification method, maybe you should rename it to 'SAR image processing' or something similar. Reply : You are right. We will change it as suggested.

Question : 4.2 Snow patch temperatures: Did you consider using external temperature measurements, e.g. from AWS or Reanalysis data, for your study? Given the narrow range of temperatures for your test site, it would have been also interesting to have temperatures available for the September image. Reply : No, we only planned to use snow temperatures. The diurnal range is really small, but we will check on the availabilty of daily data for the studied days, including the September scene. Using reanalysis can be an option. We will check the data and use it for improving the characterization of the meteorological conditions.

Question : 4.4 Wet snow patch backscattering characteristics: On page 8, lines 4-5, you state that 'Figure 9b shows that at HH polarization a weak positive correlation exists...'. I cannot see any correlation in the figure and suggest to rephrase this sentence. Reply : The figure is 10b (there is a typo in the text) and if you remove the outlier, which is the snow patch showing a grain-size of 2mm and a backscattering of -9.0 dB, you will find a R2 = 0.23 at p < 0.14, thus not statistically significant but with an identifiable weak trend. We could delete this, but we think it might provide leads to future research.

If we calculate the average dB per grain size, the correlation becomes even clearer with an $R^2 = 0.94$ at $p < 0.15$. We will clarify the text and also the graph, by including the straight line and an indication of the outlier to exclude. However, this is also something that we could easily remove.

Question : 5.3 Classification using an object oriented algorithm: Here, you use a set of morphological filters to suppress speckle and to obtain more homogeneous regions. If the quality of your threshold-based classification suffers from the same noise characteristics, then why didn't you use that set of filters for all classifications? Reply : We avoided using too much filtering in the pixel-based classification since it relies on single pixel backscattering and preferred to only use a majority filter for visualization purposes, after the evaluation of classification quality. For the object-based approach, it was necessary to remove the noise in order to improve the segmentation process and hence filtering was conducted.

Question : Table 3: There is something seriously wrong with this table. From column 7 on, it does not make any sense. Reply : You are right. We have mixed some of the columns when organizing the table for the submission. We are now attaching the correct table.

Question : Table 4: What do you mena by prod. acc. / user acc.? Please explain the abbreviations. Answer : These are two frequently used measures in confusion matrix analysis, the producer accuracy and the user accuracy. The former measures the errors of omission (pixels correctly classified as a percentage of the total nr of pixels that belong to that class), while the later measures errors of comission (the number of correctly classified pixels compared to the total number of pixels assigned to that class).

Question : Figure 12: The legend is very hard to read, please make it bigger. If you have 4 classes in the image (white, light and dark gray, black), why do you only have 3 of them in the legend? Answer : We will enlarge the legend and make it

a single common legend for both figures. We will also add the white class with the indication of unclassified. This class shows very high values of backscattering, which in our classification approach were unclassified, since they are higher than the upper boundary of bare soil. This effect is linked to artifacts associated to relief displacement, which were not resolved even with the 5 m DEM.

Question : Figure 14b: This figure is very hard to interprete, since it looks just like Fig 14a tinted red. Maybe a zoomed-in region could provide a higher level of detail? Answer : You are right. We will provide a zoomed-in window for better visualization in the revised version of the manuscript.

4 Technical corrections: Question : page 3, lines 20-22, 'Mapping of the later...': This sentence got a bit lost, it seems. Answer : We think that this sentence is important, but we will clarify it by indicating ' Mapping of snow patches and monitoring melting patterns. . . '

Question : page 4, lines 14-16, '... geocoding of the TerraSAR-X scenes and ground.': There is something missing here. Answer : You're right. We have added ' . . .and ground truthing. '

Question : page 4, line 30, 'Pervasive moisture...': This sentence appears to be a bit out of context, maybe shift it up a bit, after 'Each of the snow pits...'. Answer : You are right. Thanks. We will move the sentence as suggested.

Question : page 8, lines 15-16, 'Given the best quality...': This sentence is a bit confusing, please rephrase. The next sentence is missing a 'the'. Answer : Right. We will change it to ' The best quality of the discrimination in the HH-polarisation scene of 12 January, when compared to the VV-scene of 13 January, led us to its selection for assessing the application of backscattering thresholds and band maths for the classification. '

Question : page 8, line 24, 'thresholds '": If you use uppercase on the other scenarios,

use it here as well. Answer : You are right. We will change it accordingly.

Question : page 8, line 29, Fig. 10: should probably be Fig 12. Answer : You are right. We will correct this in the revised version of the manuscript.

Question : page 10, line 25: 'snow patches showed rare ice layers': I suggest rephrasing to '...snow patches rarely showed ice layers' Answer : Right. Thanks.

Question : page 11, lines 26-27, 'The acquisition mode is very relevant...': I don't really understand what you mean to convey with this sentence. Answer : You are right. We will remove this sentence. It is a relic from a previous draft and we forgot it here and is not needed.

New references cited:

N. Baghdadi, C. E. Livingstone, and M. Bernier, Airborne -Band SAR Measurements of Wet Snow-Covered Areas, IEEE Transactions on Geoscience and Remote Sensing, vol. 36, no. 6, november 1998 Mi, L., Hoan, N.T., Tateishi, R., Iizuka, K., Alsaaideh, B. and Kobayashi, T. (2014) A Study on Tropical Land Cover Classification Using ALOS PALSAR 50 m Ortho-Rectified Mosaic Data. Advances in Remote Sensing, 3, 208-218. http://dx.doi.org/10.4236/ars.2014.33014 Kellndorfer, J.M., Pierce, L.E., Dobson, M.C. and Ulaby, F.T. (1998) Toward Consistent Regional-to-Global-Scale Vegetation Characterization Using Orbital SAR Systems. IEEE Transactions on Geoscience and Remote Sensing, 36, 1396-1411. http://dx.doi.org/10.1109/36.718844
 Small, D., Miranda, N., Zuberbühler, L., Schubert, A. And Meier, E., Terrain-corrected Gamma: Improved thematic land-cover retrieval for SAR with robust radiometric terrain correction. Proc. 'ESA Living Planet Symposium', Bergen, Norway 28 June – 2 July 2010 (ESA SP-686, December 2010). Woodhouse I.H. : Introduction to Microwave Remote Sensing. Taylor and Francis, 2006, 284-285

[Figure]

| Snow patch | Backscattering (dB) | | | | Slope (°) | Aspect | SWE (cm) | Surface (0-5 cm) | | | Subsurface (5-10 cm) | | |
|---|---|---|---|---|---|---|---|---|---|---|---|---|---|
| | Mean | std dev | Max | Min | | | | Grain Size (mm) | Density (kg/m3) | Ice layer | Grain Size (mm) | Density (kg/m3) | Ice layer |
| 3 | -18.2 | 1.3 | -14.1 | -22.0 | 15 | N360 | 4.7 | 3 | 472 | no | 2 | 487 | yes |
| 12 | -18.2 | 1.8 | -8.2 | -22.7 | 9 | N130 | 5.5 | 4 | 550 | no | 4 | 519 | no |
| 5 | -18.8 | 1.1 | -14.9 | -23.0 | 20 | N10 | 4.7 | 3 | 472 | no | 4 | 487 | no |
| 7 | -18.8 | 1.4 | -14.8 | -23.5 | 8 | N290 | 4.6 | 3 | 456 | no | 2 | 487 | yes |
| 1 | -15.0 | 1.8 | -9.2 | -19.6 | 11 | N270 | 4.7 | 3 | 472 | no | 4 | 487 | yes |
| 2 | -9.0 | 1.4 | -4.5 | -14.9 | 34 | N270 | 5.0 | 2 | 503 | no | 4 | 550 | yes |
| 4 | -16.2 | 2.8 | -8.0 | -20.7 | 20 | N180 | 4.7 | 4 | 472 | yes | 3 | 519 | yes |
| 6 | -19.6 | 1.0 | -13.6 | -22.8 | 9 | N130 | 4.7 | 2 | 472 | yes | 1 | 550 | no |
| 8 | -17.8 | 1.8 | -4.4 | -21.6 | 6 | N170 | 5.2 | 2 | 519 | no | 2 | 519 | yes |
| 10 | -18.5 | 1.6 | -13.2 | -23.4 | 8 | N70 | 6.0 | 3 | 597 | no | 1-2 | 597 | no |
| 11 | -17.2 | 1.6 | -13.1 | -21.4 | 10 | N280 | 5.2 | 3 | 519 | yes | 3 | 519 | yes |
| 13 | -19.8 | 1.0 | -16.6 | -23.6 | 20 | N120 | 5.2 | 2 | 519 | yes | 2 | 550 | no |

**Table 3: Snow patch characteristics and backscattering in HH-Polarization (12 January 2012). Density measurements exclude the ice layers**

**Fig. 1.** Table_3_reviewed

---

## Short Comment (SC3) · 24 Oct 2016

G. Vieira

vieira@campus.ul.pt

Dear Mr Marco Jorge,

Thanks for your comments on the manuscript. Concerning your questions, please find the answers below. We will review the manuscript and clarify the issues you have raised.

Question: Section 3.3 is titled "Production and validation of snow cover maps" but does not explain how the validation was done. I think it would be beneficial to separate classification algorithms ("Production of snow cover maps") from validation. Only after the results (section 4), in section 5, is it explained that the ground truth data was divided

into classification and validation sets. I would additionally suggest explaining how the separation between classification and validation sets was performed (e.g., random?) as well as adding considerations on the representativeness of the samples (different configurations will significantly affect computed performance). Considering the small size of the reference dataset, for a minimum-bias assessment, the performance of the preferred classifier should be trained and evaluated using multiple training and validation sets (from multiple, different partitions). As is, it would be useful to have the classification and validation polygons discriminated in one or all of the results' maps; or, maybe, just remove the patches used for classifier development from those maps. Answer: We understand your comments, but prefer maintaining both production and validation under 3.3. However, we agree that the validation text, which was presented in 5. is better placed in 3.3 and we will move it there. We will also clarify the explanation of the validation procedure, e.g. random selection of the training and validation sets and explaining the validation procedure. Plotting both sets would not be feasible, since the procedure is based on the random selection of pixels and not polygons, which will show scattered in the figures and will not really add-up to the contents.

Question: The study area is quite small (< 1.5 sq. km?), yet the reference data is significantly spatially restricted; although it could be difficult to analyze snow properties for all snow patches, it is clear that wet-snow conditions are widespread – why only some patches were mapped in the field? Answer: the timing of surveying had to match by not too much time the satelitte overpass and therefore, we have selected a small area with snow patches showing different aspects. The selection was made a priori in a first field survey and only then, dataloggers were installed, snow pits dug and limits of snow patches were mapped. It is a procedure that takes time and involves using different instruments and relatively complex field logistics under difficult weather conditions in short time. A few days after the overpass, there were snow fall events that covered the terrain.

Question: Additionally, since the presented method for snow mapping involves classifying non-snow land covers, having more extensively field-mapped the non-snow classes would have enabled a more reliable performance assessment independently of the snow cover mapping. Answer: see above.

Question: My concern is that the presented values of Kappa, etc., though encouraging, do not properly convey the performance of the classifications. For example, in Fig. 15, in two instances, the snow ground-truth polygons (the northernmost and southernmost polygons) are much smaller than the SAR-image derived snow patches they overlap. Do those (red) polygons represent the actual extent of the snow patches? If so, it means that the overmapping for the snow class is much more significant than the performance measures suggest, and thus actual performance is lower than the computed performance; i.e., the geometry and distribution of the ground-truth areas would have been a strong determinant of measured performance. If not, what was the rationale for mapping only a portion of the snow patch? Answer: Small snow patches with well-defined boundaries were fully delineated with the DGPS surveying. The two snowpatches which you mention are large ones and they were only mapped close to the sites where we have installed the dataloggers. In cases where too much slush was present, we excluded the slush from the snowpatch boundary, since in some sites close to valley floors there was really more water than snow already. In synthesis, the results do not show overmapping in the two cases you have pinpointed and our knowledge of the terrains indicates that the mapping results agree with the snow patch extent, although we cannot quantify it.

Question: It would be more effective to describe the used statistics (evaluation of the different polarizations for land cover class discrimination; comparison of the classification algorithms; automated classification evaluation) under methodology. Currently, they are essentially referred for the first time in or after the results section. Answer: That could have been an approach. However, reviewer #1 considered the manuscript well-structure and we prefer to keep it as is, since the reader becomes aware of the rationale behind the application of the different methods while reading the manuscript.

If needed, we can also easily accommodate such a change.

Question: In line with a comment from reviewer #1, section 3.2 deals with data and data (pre)processing, not with image classification as suggested by the respective title; ideally, there would be a correspondence between the 3 items highlighted in the text right after section 3 header (Methodology), and section 3 level-2 headers. Answer: We will change it following the suggestions of reviewer #1.

Question: Section 5 is composed of results and thus should be under the results section (section 4). The method descriptions under section 5 would move to the methodology section. Answer: You are right. We will include section 5 under results.

———————————————

---

## Referee Comment (RC2) · J. Yackel (Referee) · 4 Nov 2016

Overall comments:

This paper utilized very high resolution TerraSAR-X imagery for maritime snow patch mapping in the Maritime Antarctic. The manuscript is clearly written, organized and detail-oriented. Several SAR classification techniques have been tested to identify snow patch in summer. SAR imagery from winter and field measurement were used as ground truth. Authors mentioned different accuracy from tested techniques to identify wet snow patches. However, there are concerns regarding terrain and incidence angle corrections. To utilize this method for operational monitoring these concerns should be resolved. Moreover, the authors mention about misclassification of wet soil with wet

snow patch. Proposed method has the potential for more accurate classification by adding classes such as wet soil, bare soil etc in classification. It would be nice to see a confusion matrix with all these categories for improved classification accuracy.

Specific comments:

What is meant by wet snow? What is the moisture content by volume? This is important in terms of microwave signature that varies with moisture content in snow. Is this method meant to identify snow patch regardless of wetness (e.g. saturated, 5% moisture etc.)?

Page 2, line 1-2 'In mountain terrain. . ..." Reference is needed to support the statement. And, why it is difficult? What are the constrains?

Page 4, section 3.1 When and how the grain size was measured? Considering the high temperature fluctuation from Fig 7, grain size will be different as well depending on the time of measurement. I think, Fig 10 would have better agreement if those measurements were coinciding.

Page 5, section 3.2 How this technique will be same/different for descending passes?

Page 6, line 27 The timing of snow temperature measurement was shown in GMT. To have a better idea about all of these dataset, all time should be mentioned in a single unit (UTC/GMT/local time: choose any and be consistence).

Page 7, line 6-7 Due to diurnal effect, backscatter from HH polarization will be varied in ascending and descending pass. How this effect was considered? Incidence angle has a significant effect on microwave backscatter. Images from Jan 12 has very large incidence angle in comparison to other two images. How us incidence angle dependency on backscatter addressed?

Page 7, Line 8 'The summer HH polarization scene showed best separability . . .' Why HH worked better than VV?

Page 8, section 5.1 How these thresholds will change with different passes/polarizations/incidence angles?

Page 10, Line 10 Overall accuracy for the classification is promising. However, looking at Fig 15, it seems ground truth polygons are not perfectly overlapped with extent from SAR images in most cases, that questions the actual performance of the approach.

Page 10, line 15 and page 11, line 19-21 'The only issue arise in classification. . . .' How can this issue be resolved?

Page 17, Table 1 Acquisition time for SAR images in local time (instead of UTC) would help to correlate the temperature during acquisition from Fig 7. As water content in snow is one of the major determinant of microwave backscatter (both HH and VV), therefore local temperature should be considered while calculating any threshold for wet snow.

Page 11, line 24 'Radar'.. should be radar

Page 21, Figure 3: 'aquisitions'. . . should be 'acquisitions'. 'analisys'. . . should be 'analysis'

---

## Author Comment (AC1) · 2 Dec 2016

Dear Dr Christian Haas, Please find below the original questions, as well as the answers to the two referees and to the public. In the end of the file, we have included the new version of the manuscript with the corrected figures, including a track of the changes we have made. We have followed the detailed comments and made several changes, which we think have contributed to improve the manuscript. We hope that it is suitable for publication in The Cryosphere. Looking forward to hear from you. Thanks and best wishes, Carla Mora, Juan Javier Jímenez, Pedro Pina, João Catalão and Gonçalo Vieira

[Figure]

Please also note the supplement to this comment:
http://www.the-cryosphere-discuss.net/tc-2016-190/tc-2016-190-AC1-supplement.pdf

---

## Author Response (AR1)

**Evaluation of single-band snow patch mapping using high resolution microwave remote sensing: an application to the Maritime Antarctic**

2 December 2016

Dear Dr Christian Haas,

Please find below the original questions, as well as the answers to the two referees and to the public. In the end of the file, we have included the new version of the manuscript with the corrected figures, including a track of the changes we have made. We have followed the detailed comments and made several changes, which we think have contributed to improve the manuscript. We hope that it is suitable for publication in The Cryosphere.

Looking forward to hear from you.

Thanks and best wishes,

Carla Mora, Juan Javier Jímenez, Pedro Pina, João Catalão and Gonçalo Vieira

**Evaluation of single-band snow patch mapping using high resolution microwave remote sensing: an application to the Maritime Antarctic**

**1 – Comments from referees/public**

**Anonymous Referee #1**

**1 General comments:**

The manuscript presents an approach to map snow patches in the Maritime Arctic using high-resolution SAR data. The authors examine different classification approaches and have conducted an extensive field campaign in order to evaluate the obtained results.They find a SVM-based classification approach suitable for mapping wet snow patches within their study region. The paper is clearly structured, the scientific methods are described in detail and the results are well presented. Nevertheless, the following points need to be addressed by the authors.

**2 Specific comments:**

There is an incidence angle dependency of the backscattered signal. Depending on the local incidence angle of your terrain (SAR scene incidence angle + terrain slope), this can become significant. You cite this effect in the introduction, and you also observe it at steeper terrain slopes, where your classification fails. If your method is intended for wider use (and in your abstract, you mention a possible operational application), how can you handle incidence angle dependency?

You found the HH scene to be better suited than the VV scene acquired on the following day. Do you have an explanation for this observation?

The water bodies you observed show very low backscatter. How would your classification approach handle wind-roughened water which can become very bright due to Bragg scattering?

1 Introduction:

In line 24-25, you state that "Most applications have been developed for regional scale mapping, but for higher resolution approaches they lack quality." This is a very strong remark, please elaborate on that.

3.1 Field characterization of the snow cover:

First of all, I think it is a very good idea to comment on the failed temperature measurements and to give a detailed analysis of the possible cause. Still, I am missing a description of the other methods of measurement. How did you measure grain size and how do you define grain size in the first place? How did you measure snow density?

3.2 SAR imagery classification:

This section does not actually describe the classification method, maybe you should rename it to "SAR image processing" or something similar.

4.2 Snow patch temperatures:

Did you consider using external temperature measurements, e.g. from AWS or Reanalysis data, for your study? Given the narrow range of temperatures for your test site, it would have been also interesting to have temperatures available for the September image.

4.4 Wet snow patch backscattering characteristics:

On page 8, lines 4-5, you state that "Figure 9b shows that at HH polarization a weak positive correlation exists...". I cannot see any correlation in the figure and suggest to rephrase this sentence.

5.3 Classification using an object oriented algorithm:

Here, you use a set of morphological filters to suppress speckle and to obtain more homogeneous regions. If the quality of your threshold-based classification suffers from the same noise characteristics, then why didn't you use that set of filters for all classifications?

**3 Tables and Figures:**

• Table 3: There is something seriously wrong with this table. From column 7 on, it does not make any sense.

• Table 4: What do you mena by prod. acc. / user acc.? Please explain the abbreviations.

• Figure 12: The legend is very hard to read, please make it bigger. If you have 4 classes in the image (white, light and dark gray, black), why do you only have 3 of them in the legend?

• Figure 14b: This figure is very hard to interprete, since it looks just like Fig 14a tinted red. Maybe a zoomed-in region could provide a higher level of detail?

**4 Technical corrections:**

• page 3, lines 20-22, "Mapping of the later...": This sentence got a bit lost, it seems.

• page 4, lines 14-16, "... geocoding of the TerraSAR-X scenes and ground.": There is something missing here.

• page 4, line 30, "Pervasive moisture...": This sentence appears to be a bit out of context, maybe shift it up a bit, after "Each of the snow pits...".

• page 8, lines 15-16, "Given the best quality...": This sentence is a bit confusing, please rephrase. The next sentence is missing a "the".

• page 8, line 24, "thresholds b": If you use uppercase on the other scenarios, use it here as well.

• page 8, line 29, Fig. 10: should probably be Fig 12.

• page 10, line 25: "snow patches showed rare ice layers": I suggest rephrasing to "...snow patches rarely showed ice layers"

• page 11, lines 26-27, "The acquisition mode is very relevant...": I don't really understand what you mean to convey with this sentence.
* * *
**Marco G. Jorge**

mjorge@sfu.ca

This is a very interesting and useful manuscript that highlights the potential of very high resolution, X-band SAR imagery for large cartographic scale, snow cover mapping. Notwithstanding, I find the manuscript's structure somewhat ineffective, particu- larly with regard to method description. Relatedly, the methodology used to evaluate the performance of the classifiers is not well explained and seems inadequate (does not translate actual performance). I hope that the following comments come across to the authors as constructive in intention; some of the issues I highlight below are issues that I deal with in my own research and have particular interest on.

Section 3.3 is titled "Production and validation of snow cover maps" but does not ex- plain how the validation was done. I think it would be beneficial to separate classification algorithms ("Production of snow cover maps") from validation. Only after the results (section 4), in section 5, is it explained that the ground truth data was divided into classification and validation sets. I would additionally suggest explaining how the separation between classification and validation sets was performed (e.g., random?) as well as adding considerations on the representativeness of the samples (different configurations will significantly affect computed performance). Considering the small size of the reference dataset, for a minimum-bias assessment, the performance of the preferred classifier should be trained and evaluated using multiple training and validation sets (from multiple, different partitions). As is, it would be useful to have the classification and validation polygons discriminated in one or all of the results' maps; or, maybe, just remove the patches used for classifier development from those maps.

The study area is quite small (< 1.5 sq. km?), yet the reference data is significantly spatially restricted; although it could be difficult to analyze snow properties for all snow patches, it is clear that wet-snow conditions are widespread – why only some patches were mapped in the field? Additionally, since the presented method for snow mapping involves classifying non-snow land covers, having more extensively field-mapped the non-snow classes would have enabled a more reliable performance assessment independently of the snow cover mapping. My concern is that the presented values of Kappa, etc., though encouraging, do not properly convey the performance of the classifications. For example, in Fig. 15, in two instances, the snow ground-truth polygons (the northernmost and southernmost polygons) are much smaller than the SAR-image derived snow patches they overlap. Do those (red) polygons represent the actual extent of the snow patches? If so, it means that the overmapping for the snow class is much more significant than the performance measures suggest, and thus actual performance is lower than the computed performance; i.e., the geometry and distribution of the ground-truth areas would have been a strong determinant of measured performance. If not, what was the rationale for mapping only a portion of the snow patch?

It would be more effective to describe the used statistics (evaluation of the different polarizations for land cover class discrimination; comparison of the classification algorithms; automated classification evaluation) under methodology. Currently, they are essentially referred for the first time in or after the results section.

In line with a comment from reviewer #1, section 3.2 deals with data and data (pre)processing, not with image classification as suggested by the respective title; ideally, there would be a correspondence between the 3 items highlighted in the text right after section 3 header (Methodology), and section 3 level-2 headers. Section 5 is composed of results and thus should be under the results section (section 4). The method descriptions under section 5 would move to the methodology section.
* * *
**J. Yackel (Referee)**

 yackel@ucalgary.ca

Overall comments: This paper utilized very high resolution TerraSAR-X imagery for maritime snow patch mapping in the Maritime Antarctic. The manuscript is clearly written, organized and detail-oriented. Several SAR classification techniques have been tested to identify snow patch in summer. SAR imagery

from winter and field measurement were used as ground truth. Authors mentioned different accuracy from tested techniques to identify wet snow patches. However, there are concerns regarding terrain and incidence angle corrections. To utilize this method for operational monitoring these concerns should be resolved. Moreover, the authors mention about misclassification of wet soil with wet snow patch. Proposed method has the potential for more accurate classification by adding classes such as wet soil, bare soil etc in classification. It would be nice to see a confusion matrix with all these categories for improved classification accuracy. Specific comments: What is meant by wet snow? What is the moisture content by volume? This is important in terms of microwave signature that varies with moisture content in snow. Is this method meant to identify snow patch regardless of wetness (e.g. saturated, 5% moisture etc.)? Page 2, line 1-2 'In mountain terrain. . ..'' Reference is needed to support the statement. And, why it is difficult? What are the constrains? Page 4, section 3.1 When and how the grain size was measured? Considering the high temperature fluctuation from Fig 7, grain size will be different as well depending on the time of measurement. I think, Fig 10 would have better agreement if those measurements were coinciding. Page 5, section 3.2 How this technique will be same/different for descending passes? Page 6, line 27 The timing of snow temperature measurement was shown in GMT. To have a better idea about all of these dataset, all time should be mentioned in a single unit (UTC/GMT/local time: choose any and be consistence). Page 7, line 6-7 Due to diurnal effect, backscatter from HH polarization will be varied in ascending and descending pass. How this effect was considered? Incidence angle has a significant effect on microwave backscatter. Images from Jan 12 has very large incidence angle in comparison to other two images. How us incidence angle dependency on backscatter addressed? Page 7, Line 8 'The summer HH polarization scene showed best separability . . .' Why HH worked better than VV?

Page 8, section 5.1 How these thresholds will change with different passes/polarizations/incidence angles? Page 10, Line 10 Overall accuracy for the classification is promising. However, looking at Fig 15, it seems ground truth polygons are not perfectly overlapped with extent from SAR images in most cases, that questions the actual performance of the approach. Page 10, line 15 and page 11, line 19-21 'The only issue arise in classification. . ..' How can this issue be resolved? Page 17, Table 1 Acquisition time for SAR images in local time (instead of UTC) would help to correlate the temperature during acquisition from Fig 7. As water content in snow is one of the major determinant of microwave backscatter (both HH and VV), therefore local temperature should be considered while calculating any threshold for wet snow. Page 11, line 24 'Radar'.. should be radar Page 21, Figure 3: 'aquisitions'. . . should be 'acquisitions'. 'analisys'. . . should be 'analysis

**2- Author's response**

**Reply to Anonymous Referee #1**

Thank you very much for the questions and remarks that we believe contribute to clarify several parts of the manuscript. Below, we present a detailed answer to the questions you have raised concerning the text and tables.We have clarified the manuscript accordingly, and have also included your technical corrections.

Question: There is an incidence angle dependency of the backscattered signal. Depending on the local incidence angle of your terrain (SAR scene incidence angle + terrain slope), this can become significant. You cite this effect in the introduction, and you also observe it at steeper terrain slopes, where your classification fails. If your method is intended for wider use (and in your abstract, you mention a possible operational application), how can you handle incidence angle dependency?

Reply : This is a significant issue that does include limitations linked to the difficulties on accurately modelling the backscattering, but that we think could be mitigated with improved digital elevation models (e.g. better accuracy and better resolution). An approach could be by using UAV-based aerial photo surveying and DSM generation. However, such models will never be perfect due to temporal changes in snow accumulation patterns inducing varying local snow morphologies. Slope have been widely studied to introduce geometric corrections (e.g. Mi et al, 2014; Small et al, 2010), but we have adopted a simpler but robust approach through Range Doppler Terrain Correction, taking into account the advantage of a 5 m DEM. However, as we show, some incidence angle + slope relationships will remain difficult to resolve. In the procedure that we have applied, the incidence angle and terrain slope are both considered in the absolute radiometric calibration to sigma nougth in ESA-SNAP software (Kellndorfer et al, 1998), and in the subsequent phase of Range Doppler Terrain Correction. Using imagery showing multiple SAR incident angle backscattering responses, and both ascending and descending passes, would be the best approach to infer a more complete radiometric perfomance of the terrain signal and is a recommended practice to regionalise the results. Unfortunately, in this case only two scenes were available (HH with 45.626 incident angle and VV with 29.875 incident angle). The original plan was to have more imagery, but not all acquisitions were guaranteed. We have improved this discussion in the manuscript (see p. 11 – end of discussion) and also indicated that the paper is a step towards the operational implementation of the method.

Question: You found the HH scene to be better suited than the VV scene acquired on the following day. Do you have an explanation for this observation?

Reply: We have found a similar behaviour when using Envisat ASAR imagery for Deception Island (Mora et al 2013), as well as other authors, such as Baghdadi et al., 1998 using polarimetric SAR data at C-band for the classification of land covers (open areas, lakes ice, and forests, all covered with wet snow) and they have also concluded that HH-polarisation is better than VV-polarisation. The backscattering behavior is dependent on the dryness of the snow, on the incident angle and on the roughness of the surface. For classification purposes the most important issue is the separability between classes and in this case, it seems that HH is more appropriate to separate between water, bare soil and wet snow. Additionally, VV polarisation is more sensitive to water roughness changes. In the case of our scenes, the HH scene shows a higher incidence angle, which improves resolution in a terrain with an irregular topography (Woodhouse, I., 2006), such as the study area. We changed the manuscript by adding this discussion (p. 11, end of 1st paragraph).

Question : The water bodies you observed show very low backscatter. How would your classification approach handle wind-roughened water which can become very bright due to Bragg scattering?

Reply : This is a very good question. In order to implement our approach operationally, the lake surfaces should be masked after an initial detection. This would pose issues where lake water levels vary very significantly, or where lakes cover a large percent of the terrain, but neither is the case in the ice-free areas of the Maritime Antarctic. So, an initial assessment of lake boundaries, either using imagery in low wind conditions, or using optical imagery, could be used to create a lake mask. We have introduced this discussion in the manuscript, in page 12, in the end of the discussion.

Question : In line 24-25, you state that "Most applications have been developed for regional scale mapping, but for higher resolution approaches they lack quality." This is a very strong remark, please elaborate on that.

Reply : You are right. We have clarified the sentence and deleted the last part. Essentially, the literature lacks published results on the use of SAR for snow mapping at very high resolution (metric).

Question : 3.1 Field characterization of the snow cover: First of all, I think it is a very good idea to comment on the failed temperature measurements and to give a detailed analysis of the possible cause. Still, I am missing a description of the other methods of measurement. How did you measure grain size and how do you define grain size in the first place? How did you measure snow density?

Reply : You are right. We have now improved the description of the snow-pit characterization. Grain-size was measured by carefully collecting small amounts of snow from each of the layers of the snow pack and by depositing them in a black tissue for contrast. They were then observed with a 10x magnifier, which allowing for measuring and describing the grain shape and size. Grain size (or crystal size) showed variability some within each layer and our descrition encompasses the mean grain-sizes, but when variability was large, we included the more frequent dimensions (i.e. 1-2 mm). Snow density was measured by carefully collecting snow from each snow layer without disturbing the density, using a metal box with a volume of 212 cm3. From each layer, 3 boxes were collected, adding up 636 cm3, which were inserted in a plastic bag and weighted using a digital spring scale, and mass converted to density.

Question : 3.2 SAR imagery classification: This section does not actually describe the classification method, maybe you should rename it to "SAR image processing" or something similar.

Reply : You are right. We changed it as suggested.

Question : 4.2 Snow patch temperatures: Did you consider using external temperature measurements, e.g. from AWS or Reanalysis data, for your study? Given the narrow range of temperatures for your test site, it would have been also interesting to have temperatures available for the September image.

Reply: No, originally we only planned to use snow temperatures. We have obtained mean daily temperatures for Bellingshausen station, located close to the study area and we have prepared a new figure (Fig. 8). The data shows that the 26 September still shows below 0 ℃ mean air temperatures, while the January scenes show temperatures above 0 ℃, confirming our original interpretation, but now providing quantitative data.

Question : 4.4 Wet snow patch backscattering characteristics: On page 8, lines 4-5, you state that "Figure 9b shows that at HH polarization a weak positive correlation exists...". I cannot see any correlation in the figure and suggest to rephrase this sentence.

Reply: The figure is 10b (there is a typo in the text) and if you remove the outlier, which is the snow patch showing a grain-size of 2mm and a backscattering of -9.0 dB, you will find a $R2 = 0.23$ at $p < 0.14$, thus not statistically significant but with an identifiable weak trend. We could delete this, but we think it might provide leads to future research. If we calculate the average dB per grain size, the correlation becomes even clearer with an $R2 = 0.94$ at $p < 0.15$. We have now clarified the text (p. 8) and also the figure 10b, by including the straight line and an indication of the outlier to exclude. However, this is also something that we can easily remove if needed to.

Question : 5.3 Classification using an object oriented algorithm: Here, you use a set of morphological filters to suppress speckle and to obtain more homogeneous regions. If the quality of your threshold-based classification suffers from the same noise characteristics, then why didn't you use that set of filters for all classifications?

Reply : We avoided using too much filtering in the pixel-based classification since it relies on single pixel backscattering and preferred to only use a majority filter for visualization purposes, after the evaluation of classification quality. For the object-based approach, it was necessary to remove the noise in order to improve the segmentation process and hence filtering was conducted.

Question : Table 3: There is something seriously wrong with this table. From column 7 on, it does not make any sense.

Reply : You are right. We have mixed some of the columns when organizing the table for the submission. We have now included the correct table.

Question : Table 4: What do you mean by prod. acc. / user acc.? Please explain the abbreviations.

Answer : These are two frequently used measures in confusion matrix analysis, the producer accuracy and the user accuracy. The former measures the errors of omission (pixels correctly classified as a percentage of the total nr of pixels that belong to that class), while the later measures errors of comission (the number of correctly classified pixels compared to the total number of pixels assigned to that class). Since we do not discuss these measures in the text, altough they provide support to the quality of the results and this should be included in Table 4, we decided to add this explanation in the table caption.

Question : Figure 12: The legend is very hard to read, please make it bigger. If you have 4 classes in the image (white, light and dark gray, black), why do you only have 3 of them in the legend?

Answer : We have enlarged the legend and made it a single common legend for both figures. We have also added the white class with the indication of "unclassified". This class shows very high values of backscattering, which in our classification approach were unclassified, since they are higher than the upper boundary of bare soil. This effect is linked to artifacts associated to relief displacement.

Question : Figure 14b: This figure is very hard to interpret, since it looks just like Fig 14a tinted red. Maybe a zoomed-in region could provide a higher level of detail?

Answer : You are right. We have now provided a zoomed-in window for better visualization.

4 Technical corrections:

Question : page 3, lines 20-22, "Mapping of the later...": This sentence got a bit lost, it seems.

Answer : We think that this sentence is important, since it targets partly at the potential readers and application, but have clarified it by indicating « Mapping of snow patches and monitoring melting patterns… »

Question : page 4, lines 14-16, "... geocoding of the TerraSAR-X scenes and ground.": There is something missing here.

Answer : You're right. We have added « …and ground truthing. »

Question : page 4, line 30, "Pervasive moisture...": This sentence appears to be a bit out of context, maybe shift it up a bit, after "Each of the snow pits...".

Answer : You are right. Thanks. We will move the sentence as suggested.

Question : page 8, lines 15-16, "Given the best quality...": This sentence is a bit confusing, please rephrase. The next sentence is missing a "the".

Answer : Right. We have changed it to « The best quality of the discrimination in the HH-polarisation scene of 12 January, when compared to the VV-scene of 13 January, led us to its selection for assessing the application of backscattering thresholds and band maths for the classification. »

Question : page 8, line 24, "thresholds b": If you use uppercase on the other scenarios, use it here as well.

Answer : You are right. We have changed it accordingly.

Question : page 8, line 29, Fig. 10: should probably be Fig 12.

Answer : You are right. It is now corrected.

Question : page 10, line 25: "snow patches showed rare ice layers": I suggest rephrasing to "...snow patches rarely showed ice layers"

Answer : Right. Thanks.

Question : page 11, lines 26-27, "The acquisition mode is very relevant...": I don't really understand what you mean to convey with this sentence.

Answer : You are right. We have removed this sentence. It is a relic from a previous draft and we forgot it here and is not needed.

-------------------------------------------------------------------------------------------------------------------- -

**Reply to Marco Jorge**

**Dear Mr Marco Jorge,**

Thanks for your comments on the manuscript. Concerning your questions, please find the answers below. We will review the manuscript and clarify the issues you have raised.

Question: Section 3.3 is titled "Production and validation of snow cover maps" but does not explain how the validation was done. I think it would be beneficial to separate classification algorithms ("Production of snow cover maps") from validation. Only after the results (section 4), in section 5, is it explained that the ground truth data was divided into classification and validation sets. I would additionally suggest explaining how the separation between classification and validation sets was performed (e.g., random?) as well as adding considerations on the representativeness of the samples (different configurations will significantly affect computed performance). Considering the small size of the reference dataset, for a minimum-bias assessment, the performance of the preferred classifier should be trained and evaluated using multiple training and validation sets (from multiple, different partitions). As is, it would be useful to have the classification and validation polygons discriminated in one or all of the results' maps; or, maybe, just remove the patches used for classifier development from those maps.
Answer: We understand your comments, but prefer maintaining both production and validation under 3.3. However, we agree that the validation text, which was presented in 5. is better placed in 3.3 and we will move it there. We will also clarify the explanation of the validation procedure, e.g. random selection of the training and validation sets and explaining the validation procedure. Plotting both sets would not be feasible, since the procedure is based on the random selection of pixels and not polygons, which will show scattered in the figures and will not really add-up to the contents.

Question: The study area is quite small (< 1.5 sq. km?), yet the reference data is significantly spatially restricted; although it could be difficult to analyze snow properties for all snow patches, it is clear that wet-snow conditions are widespread – why only some patches were mapped in the field?
Answer: the timing of surveying had to match by not too much time the satelitte overpass and therefore, we have selected a small area with snow patches showing different aspects. The selection was made a priori in a first field survey and only then, dataloggers were installed, snow pits dug and limits of snow patches were mapped. It is a procedure that takes time and involves using different instruments and relatively complex field logistics under difficult weather conditions in short time. A few days after the overpass, there were snow fall events that covered the terrain.

Question: Additionally, since the presented method for snow mapping involves classifying non-snow land covers, having more extensively field-mapped the non-snow classes would have enabled a more reliable performance assessment independently of the snow cover mapping.
Answer: see above.

Question: My concern is that the presented values of Kappa, etc., though encouraging, do not properly convey the performance of the classifications. For example, in Fig. 15, in two instances, the snow ground-truth polygons (the northernmost and southernmost polygons) are much smaller than the SAR-image derived snow patches they overlap. Do those (red) polygons represent the actual extent of the snow patches? If so, it means that the overmapping for the snow class is much more significant than the performance measures suggest, and thus actual performance is lower than the computed performance; i.e., the geometry and distribution of the ground-truth areas would have been a strong determinant of measured performance. If not, what was the rationale for mapping only a portion of the snow patch?

Answer: Small snow patches with well-defined boundaries were fully delineated with the DGPS surveying. The two snowpatches which you mention are large ones and they were only mapped close to the sites where we have installed the dataloggers. In cases where too much slush was present, we excluded the slush from the snowpatch boundary, since in some sites close to valley floors there was really more water than snow already. In synthesis, the results do not show overmapping in the two cases you have pinpointed and our knowledge of the terrains indicates that the mapping results agree with the snow patch extent, although we cannot quantify it.

We have added the following sentence to section 3.1: In cases where slush was present, we excluded the area from the snowpatch boundary surveying and in very large snow patches, only partial ground truthing was conducted.

Question: It would be more effective to describe the used statistics (evaluation of the different polarizations for land cover class discrimination; comparison of the classification algorithms; automated classification evaluation) under methodology. Currently, they are essentially referred for the first time in or after the results section.

Answer: That could have been an approach. However, reviewer #1 considered the manuscript well-structured and we prefer to keep it as is, since the reader becomes aware of the rationale behind the application of the different methods while reading the manuscript. If needed, we can also easily accommodate such a change.

Question: In line with a comment from reviewer #1, section 3.2 deals with data and data (pre)processing, not with image classification as suggested by the respective title; ideally, there would be a correspondence between the 3 items highlighted in the text right after section 3 header (Methodology), and section 3 level-2 headers.

Answer: We have changed the manuscript following the suggestions of reviewer #1.

Question: Section 5 is composed of results and thus should be under the results section (section 4). The method descriptions under section 5 would move to the methodology section.

Answer: You are right. We have included section 5 under results.
* * *
**Reply to J. Yackel (Referee #2)**

Dear Professor John Yackel,

Thank you very much for your very good comments and ideas, which we have answered below. We agree with your observations and it is true our approach shows some limitations, which can be overcome with future research designs, new field campaigns and some more luck with image acquisition plans. Our original plan was to have more imagery, but unfortunately this was not possible due to operational reasons from the image provider. We have also faced some constraints with the field survey. However, our very high resolution approach, with a significant number of sampled snow patches is, to our knowledge, novel, at least in the Maritime Antarctic and we think the methodology demonstrates the high potential for application of TerraSAR-X for high resolution snow monitoring, especially during the melting season. This could be a very significant step for various disciplines studying the ice-free areas of the Maritime Antarctica, where summer snow melt plays a major ecological, hydrological and geomorphological role. Thanks and our best wishes,

Carla Mora and co-authors.

Overall comments: This paper utilized very high resolution TerraSAR-X imagery for maritime snow patch mapping in the Maritime Antarctic. The manuscript is clearly written, organized and detail-oriented. Several SAR classification techniques have been tested to identify snow patch in summer. SAR imagery from winter and field measurement were used as ground truth. Authors mentioned different accuracy from tested techniques to identify wet snow patches. However, there are concerns regarding terrain and incidence angle corrections. Moreover, the authors mention about misclassification of wet soil with wet snow patch.

Question: Proposed method has the potential for more accurate classification by adding classes such as wet soil, bare soil etc in classification. It would be nice to see a confusion matrix with all these categories for improved classification accuracy.

Answer: You are correct. However, the experimental setting was designed to evaluate the possibility to detect with high spatial resolution the patterns of snow melt during the summer. Originally we were expecting more intersnowpatch diversity in snow conditions and not so extreme melt conditions all over. We did not collect information on soil wetness and this limits the approach which you propose.

Specific comments:

Question: What is meant by wet snow? What is the moisture content by volume? This is important in terms of microwave signature that varies with moisture content in snow. Is this method meant to identify snow patch regardless of wetness (e.g. saturated, 5% moisture etc.)?

Answer: The moisture content by volume was not measured and yes, the method is intended to map snow patch extent and aims at mapping snow melt patterns by using multitemporal imagery. This was one of the objectives of the field season. However, some of the scenes, which we had requested, were not collected

due to operational reasons. We added a sentence in section 3.1 explaining that snow moisture content by volume was not measured.

Question: Page 2, line 1-2 'In mountain terrain. . ..'' Reference is needed to support the statement. And, why it is difficult? What are the constrains?
Answer: The difficulty arises from terrain shadowing and lay-over effects. We have added a reference to Rees, W. S. and Steel, M.: Radar backscatter coefficients and snow detectability for upland terrain in Scotland. International Journal of Remote Sensing, 22, 15, 3015-3026, 2001.

Question: Page 4, section 3.1 When and how the grain size was measured? Considering the high temperature fluctuation from Fig 7, grain size will be different as well depending on the time of measurement. I think, Fig. 10 would have better agreement if those measurements were coinciding.
Answer: The high temperature variation for the daily maxima as shown in the graph is an artifact related to the overheating of the minilogger package, which was just below the snow surface and at times, exposed. The snow itself melted a few centimeters each day, as seen from the resurfacing of the miniloggers. It is possible that grain size has varied between the time of field measurements, made in the 11 January and the image acquisition time, but such changes should have been small. We agree, however, that possibly, there would be a better agreement with fig. 10. For logistical reasons in the field, the measurement date was selected as a good approach between the different acquisitions that were originally planned. The results aiming at snow patch boundary detection are, however, not affected by the lack of agreement in fig. 10. We have added a reference to the date of the snow pit surveying in the beginning of section 3.1. The limitations of the snow temperature data were already explained in the original manuscript.

In what respects to the measurement of grain size, it was done by collecting small amounts of snow from each of the layers of the snow pack and by depositing them in a black tissue for contrast. The description was done using a mm graduated ruller and a 10x magnifier lens. We have included the information arising from these questions in the new version of the manuscript.

Question: Page 5, section 3.2 How this technique will be same/different for descending passes?
Answer: The techique is the same. The method considers the incidence angle and the terrain slope in the absolute radiometric calibration to sigma nougth using ESA-SNAP software (Kellndorfer et al, 1998), and in the subsequent phase of Range Doppler Terrain Correction. Lay-over and shadowing effects occuppy a minimal area in the study sector and have not been accounted for. For more complex terrain a masking substraction would be needed to eliminate those effects. For application in operational mode, a standard single pass direction and angle should be used.

Question: Page 6, line 27 The timing of snow temperature measurement was shown in GMT. To have a better idea about all of these dataset, all time should be mentioned in a single unit (UTC/GMT/local time: choose any and be consistence).
Answer: You are right. We have changed it to UTC.

Question: Page 7, line 6-7 Due to diurnal effect, backscatter from HH polarization will be varied in ascending and descending pass. How this effect was considered?

Answer: We did not consider the above-mentioned effect and assumed that the radiometric calibration and terrain correction algorithm resolve it.

Question: Incidence angle has a significant effect on microwave backscatter. Images from Jan 12 have very large incidence angle in comparison to other two images. How is incidence angle dependency on backscatter addressed?

Answer: This question was also raised by referee #1. Incidence angle and terrain slope are both considered in the absolute radiometric calibration to sigma nought for TerraSAR X aquisitions, generated by the operator in ESA-SNAP software (Kellndorfer et al, 1998), and in the geometric georeferencing and terrain correction (Range Doppler Terrain Correction), subsequent phase into the image correction process. Analysing the responses of several SAR incident angle responses in the backscattering is the optimal approach to assess a more complete radiometric perfomance of the terrain signal and is a recommended practice to widen the study. Unfortunately, in this case, due to operational issues and limited field time, only two images were available in the period with field observations. The slope effects and incident angle approach are used to introduce geometric corrections (Mi et al, 2014; Small et al, 2010). In our case, a simple but robust approach has been applied through Range Doppler Terrain Correction, taking in account the advantage of a good 5 m DEM.

Question: Page 7, Line 8 'The summer HH polarization scene showed best separability . . .' Why HH worked better than VV?

Answer: We have found a similar behaviour when using Envisat ASAR imagery for Deception Island (Mora et al 2013), as well as other authors, such as Baghdadi et al., 1998 using polarimetric SAR data at C-band for the classification of land covers (open areas, lakes ice, and forests, all covered with wet snow) and they have also concluded that HH-polarisation is better than VV-polarisation. The backscattering behavior is dependent on the dryness of the snow, on the incident angle and on the roughness of the surface. For classification purposes the most important issue is the separability between classes and in this case, it seems that HH is more appropriate to separate between water, bare soil and wet snow. Additionally, VV polarisation is more sensitive to water roughness changes. In the case of our scenes, the HH scene shows a higher incidence angle, which improves resolution in a terrain with an irregular topography (Woodhouse, I., 2006), such as the study area. We have changed the manuscript by adding this discussion.

Question: Page 8, section 5.1 How these thresholds will change with different passes/polarizations/incidence angles?

Answer: There may be small differences in the thresholds and this is a field we are currently exploring in the framework of a PhD dissertation of one of the co-authors. An analysis of the literature on microwave remote sensing of snow shows that backscattering thresholds are widely used and we think that our approach, with a transparent presentation of the characteristics of the imagery and field data is a very valid contribution to the body of literature on the subject.

Question: Page 10, Line 10 Overall accuracy for the classification is promising. However, looking at Fig 15, it seems ground truth polygons are not perfectly overlapped with extent from SAR images in most cases, that questions the actual performance of the approach.

Answer: The vast majority of the snow areas mapped in the field as ground truth are classified as snow with the algorithm. In some sites, however, the areas classified as snow are much larger than the boundaries of the ground truthing. This is because we didn't sample the whole snowpatch. This could be due to significant aspect variation within the same snowpatch (e.g. crossing a valley floor) or due to the very large size of the snowpatch (e.g. in the SE corner of the map). We have also kept slush sections out of the sampled snow patches. Actually the overall performance is in good agreement with the modelled snow distribution. This question has also been raised in the public discussion and we have clarified the sampling procedure and validation.

Question: Page 10, line 15 and page 11, line 19-21 'The only issue arise in classification. . ..' How can this issue be resolved?

Answer: a possible solution could be by using a combined approach with the ascending and the descending pass. We have added a sentence to the text reflecting this.

Question: Page 17, Table 1 Acquisition time for SAR images in local time (instead of UTC) would help to correlate the temperature during acquisition from Fig 7. As water content in snow is one of the major determinant of microwave backscatter (both HH and VV), therefore local temperature should be considered while calculating any threshold for wet snow.

Answer: The aim of figure 7 was to show that snow is close to the melting point. As we have discussed in the methodology, the temperature shown in figure t is not the real temperature of the snow it self, but rather resulting from the heating of the small plastic container of the logger, which absorbs radiation and heats up. Unfortunately, this temperature data cannot be used for any correlation, but only to show that snow is close or at the melting point, and not at subfreezing temperatures.

Page 11, line 24 'Radar'.. should be radar

Answer: Right. We have corrected it across the manuscript.

Page 21, Figure 3: 'aquisitions'. . . should be 'acquisitions'. 'analisys'. . . should be 'analysis

Answer: Thanks. We have corrected it.

[revised manuscript text omitted]

---

## Author Response (AR2)

Lisbon, 24 December 2016

Dear Professor Christian Haas,

Thank you very much for your comments and review. We agree with your points and have changed the manuscript as you suggested.

Please find below our answers, which we hope that will make the manuscript fit for publication in The Cryosphere.

Best regards and Season Greetings,

Carla Mora and co-authors

▪▪▪▪▪▪▪▪▪▪▪▪▪▪▪▪▪▪▪▪▪▪▪▪▪▪▪▪▪▪▪▪▪▪▪▪▪▪▪▪▪▪▪▪▪▪▪▪▪▪▪▪▪▪▪▪▪▪▪▪▪

Editor Decision: Publish subject to minor revisions (Editor review) (14 Dec 2016) by Dr. Christian Haas

Comments to the Author:

Dear Authors,

Thank you for the revision of your manuscript, and for considering the reviewer's comments. I find your revisions mostly acceptable, however, I think that you should still more clearly address some of the shortcomings of your study, as mentioned by the reviewers. You answered many concerns of the third reviewer (Yackel), but you did not clearly include your responses in the manuscript.

> Reply : We acknowledge your remarks and we have now included in the manuscript the responses to the third reviewer. Please see our more detailed approach below.

Most importantly, please address the issue of variable incident angles more clearly. Despite calibration and DEM correction there is the issue of physical backscatter vs incident angle variations even for level, homogeneous surfaces. This is most pronounced for surface scattering. Therefore the question really arises if your results would be applicable to data with different incident angles. One could suspect that incidence angle differences somewhat contributed to your results, and that the role of polarization may be less important? There are too many studies showing uncritical evaluation of SAR data application for various problems, often overselling the applicability of SAR. Most of these do not pay attention to incident angle variations. In this regard, could you please also provide more explanations for the physical reasons believed to yield better results with HH than with VV in your study and others.

> Reply : The objective of this manuscript was not comparing the potential of different incidence angles, but rather to assess the quality of the SpotLight imagery for mapping snow patch boundaries at very high resolution. Therefore in our experimental design we focused in having scenes at the same dates we were on the field, rather than evaluating other (important variables). Several ordered scenes were not acquired and this has also limited the results. We think, however, that you are right with your observations and while reading again the manuscript, the lack of quality of the VV-Pol scene was probably

overemphasised. We have rewritten parts of the discussion including more information and focused on empirical data from other studies and we have made it clear that we cannot with the current data assess if the differences in quality between HH and VV are due to the incidence angle or from the polarisation itself. We have also added a line on this in the conclusions. We think that the current version respects the objectives of the manuscript and also complies with your observation relating to the uncritical evaluation and overselling SAR data.

Also please be more clear in the manuscript about diurnal variations and the effect of different acquisition times. I think the reviewer's concern is quite valid, that there will be strong diurnal variations in this season (as seen with the temperature data), and therefore backscatter will vary strongly and the applicability of your results at all times more difficult? Or do you think that the high snow wetness would buffer diurnal temperature and backscatter variations? In any case the arguments need to be better brought forward.

Reply : The diurnal temperature variations are minimal and the graphs show wrong values for the diurnal part of the day. We are sure that this is a problem with the observational set-up, but we thought it would still be worth showing the data, especially focussing on the value of the nighttime data. We have now re-emphasised this in the text, both in methods (4.2) and in the discussion and we hope it is more clear now. We have added also a sentence on the buffering related to the snow wetness. Another option is indicating with shading the window of biased temperatures in fig. 7. Inside the snow pack, the temperature should have remained close to 0 ºC.

In this regard it would also be good if you could state what the time difference between UTC and your solar time is.

Reply : Right, done. It is about -4h (59º long).  It is now added in 4.2

You make a statement about higher resolution of SAR at higher incident angles. This is true theoretically, however, the products you use have been regridded with constant pixel size, so that the resolution is the same everywhere?

Reply : we have clarified the sentence. This related to backscattering resolution.

Please consider these comments and provide responses.

Please also check for minor typos included in your revisions, and to shorten the abstract (first part).

Reply : We have checked for typos and also shortened the abstract.

Thank you and best regards

Christian Haas

[revised manuscript text omitted]